



# Modes of Antarctic tidal grounding line migration revealed by ICESat-2 laser altimetry

Bryony I. D. Freer[1,2], Oliver J. Marsh[1], Anna E. Hogg[2], Helen Amanda Fricker[3], Laurie Padman[4]

[1]British Antarctic Survey, Cambridge, CB3 0ET, UK
[2]School of Earth and Environment, University of Leeds, LS2 9JT, UK
[3]Scripps Polar Center, Scripps Institution of Oceanography, UC San Diego, California, USA
[4]Earth and Space Research, Corvallis, OR, USA

*Correspondence to*: Bryony I. D. Freer (breer90@bas.ac.uk)

**Abstract.** Short-term tidal grounding line (GL) migration in Antarctica can impact ice dynamics at the ice sheet margins and obscures assessments of long-term GL advance or retreat. However, the magnitude of tidally-induced GL migration is poorly known, and the spatial pattern and modes of variability are not well characterised. Here we develop and apply a technique that uses ICESat-2 repeat-track laser altimetry to locate the inland limit of tidal ice shelf flexure for each sampled tide, enabling the magnitude and temporal variability of tidal GL migration to be resolved. We demonstrate its application at an ice plain
north of Bungenstockrücken, in a region of the southern Ronne Ice Shelf subject to large ocean tides. We observe a 1,300 km$^2$ area of ephemeral grounding over which the GL migrates by up to 15 km between low and high tide, and identify four distinct modes of migration: "linear", "asymmetric", "threshold" and "hysteresis". The short-term movement of the GL dominates any long-term migration signal in this location, and the distribution of GL positions and modes contains information about spatial variability in the ice-bed interface. We discuss the impact of extreme tidal GL migration on ice shelf-ocean-subglacial systems
in Antarctica and make recommendations for how GLs should be more precisely defined and documented in future by the community.

## 1 Introduction

Recent mass loss from the Antarctic Ice Sheet has been attributed to changes in the floating ice shelves that fringe 74% of its
margins (Gudmundsson et al., 2019; Joughin et al., 2012) and buttress upstream grounded ice (Dupont and Alley, 2005). Accurate representation of the ice sheet in coupled Earth system models is essential to predict its evolution and the magnitude and rate of its future contribution to sea level rise. This requires knowledge of ice sheet processes and boundary conditions, including the configuration and behaviour of ice shelves. One important parameter for ice sheet monitoring and





modelling is the location of the grounding line (GL), which marks the boundary between the grounded ice sheet and floating
ice shelf (Thomas, 1979). As the ice detaches from the underlying bed, ice flow transitions from a regime dominated by
vertical shear and basal drag to that of primarily buoyancy-driven flow dominated by longitudinal stretching and lateral shear
with no basal drag (Schoof, 2007). There are high rates of basal melt at the GL as ocean water comes into contact with the
base of the ice shelf (Jenkins et al., 2006; Depoorter et al., 2013). The location of the GL and change in its position over time
is, therefore, a sensitive indicator of ice sheet stability and local environmental forcing, with sustained ice thinning causing
GL retreat, and thickening causing GL advance (Joughin et al., 2010, 2012; Dutrieux et al., 2014), and it has been identified
as an Essential Climate Variable (Bojinski et al., 2014)

Several satellite-based techniques have been used to map features in the grounding zone (GZ), the 1-10 km-wide region
spanning the GL (Vaughan, 1995), with the primary goal to monitor long-term change in GL position (Sect. 2). The GL is
typically located by identifying the inland limit of ice shelf flexure, but this is complicated by the effect of short-term sea
level variations, primarily driven by ocean tides, which can cause GL migration on hourly to daily timescales. During a
rising tide, increased buoyancy causes more of the ice shelf to lift off the bed and the GL temporarily migrates inland,
returning to its most seaward position at low tide (Brancato et al., 2020), with some slight time lag due to the viscoelastic
effects of the ice (Reeh et al., 2003). Prior satellite observations have shown that the extent of this short-term tidal GL
migration can range from a few hundred metres to several kilometres (Hogg et al., 2016; Brunt et al., 2011; Milillo et al.,
2017), with the magnitude of this migration controlled by the tide amplitude, local bed topography and the thickness and
strength of the ice. The short-term tidal GL migration signal is superimposed on top of the long-term signal of GL migration
associated with ice dynamic change, which is traditionally calculated by comparing sparsely sampled, individual
measurements of GL position through time (e.g. Rignot et al., 2014). Therefore, in order to increase confidence in
measurements of long-term GL migration rates, we must improve our knowledge of the spatially variable pattern of short-
term tidal GL migration.

Despite the importance of understanding tidal GL migration there is currently no Antarctic-wide assessment of its extent or
variability. This is mostly because there is a lack of acquired or suitable satellite data in the historical archive with sufficient
spatial and temporal resolution. We currently also have a limited understanding of the *mechanisms* of tidal ice shelf flexure
and GL migration. Various representations of tidal flexure have been modelled using elastic and viscoelastic frameworks
(Holdsworth, 1969; Vaughan, 1995; Schmeltz et al., 2002; Walker et al., 2013), some of which allow for tidal GL migration
(Sayag and Worster, 2013; Tsai and Gudmundsson, 2015) but results vary considerably and the choice of representation of
these processes can have a large impact on results of modelling studies (e.g. Mosbeux et al., 2022). Whilst some localised *in
situ* measurements of tidal flexure have been made to examine these processes (Smith, 1991; Vaughan, 1995), there are very
few satellite observations with high enough tidal sampling to test and validate these models. Furthermore, modelled
projections of future mass loss from West Antarctica are very sensitive to the amount of basal melt and ice flux at the GL



(Arthern and Williams, 2017; Goldberg et al., 2019), yet most numerical ice sheet models typically assume a fixed or slowly moving GL determined by flotation conditions alone with zero basal melt (Milillo et al., 2017; Favier et al., 2014). This is an over-simplification in areas subject to extreme tidal variability, where short-term GL migration is likely to impact both ice dynamics through rapid variations in basal shear stress, and basal melt rate through changes in cavity geometry enhancing

tidal mixing. Quantifying the extent and influence of tidal GL migration around the margin of Antarctica is valuable for these processes to be more accurately parameterised in ice sheet models.

In this study we present a new approach to observe tidal GL migration, that takes advantage of the high along-track resolution and repeat-track configuration of the Ice, Cloud and land Elevation (ICESat-2) satellite laser altimeter mission to improve temporal sampling of tidal ice flexure and GL migration. We apply the method to the ice plain north of the

Bungenstockrücken ice rise, on the southern Ronne Ice Shelf. An ice plain is defined as an area of low surface slope close to the GZ where the ice is close to flotation (Alley et al., 1989), and which can sometimes experience "ephemeral grounding" between high and low tide (Schmeltz et al., 2001; Brunt et al., 2011). We show that with ICESat-2 repeat-track laser altimetry (RTLA) it is possible to monitor the time-varying GZ structure at the Bungenstockrücken ice plain in unprecedented detail, and we identify four distinct modes of tidal GL migration. We discuss the implications both for long-

term GL monitoring and how these findings inform our process understanding of tidal flexure and GL migration mechanisms in Antarctica.

## 2 Grounding lines and their detection

### 2.1 Features of the grounding zone

The grounding zone (GZ) is a region between the seaward limit of the ice sheet and the landward limit of the ice shelf,

typically 1-10 km wide, over which the ice transitions from being in constant contact with the bed to floating in hydrostatic equilibrium with the ocean. Ice in the GZ is supported by both the hydrostatic pressure from the underlying ocean and internal stresses, and undergoes tidal flexure (Vaughan, 1995). The true GL location, Point G, refers to the point at which the ice base first detaches from the bed as it flows from the ice sheet. As a subglacial feature within the GZ, Point G can only be directly observed using ground-based radar (MacGregor et al., 2011; Catania et al., 2010) or using remotely-operated

vehicles deployed from the shelf edge or through boreholes (Schmidt et al., 2023). However, there are a number of observable surface features in the GZ that relate to the location of Point G, which can be more readily detected *in situ* or in satellite data and used as GL proxies (Fig. 1). Point F is the landward limit of tidal flexure, and is often located slightly inland of Point G (on the order of hundreds of metres) due the elastic properties of the ice (Padman et al., 2018; Rignot et al., 2011).  Point H is the seaward limit of tidal flexure and inland limit of hydrostatic equilibrium. The break-in-slope, Point $I_b$,

marks the sharp reduction in surface slope that occurs due to the abrupt change in basal shear stress across the GL, and in




typical GZs it is located slightly seaward of Point G (Schoof, 2011). However, where a GZ has an ice plain, Point $I_b$ has been observed several kilometres inland of Point G (Corr et al., 2001). We show the locations of the key GZ surface proxies as they relate to Point G for an idealised ice plain in Fig. 1(a), and in Fig. 1(b) we depict how Point F can migrate by several kilometres across the tide cycle at an ice plain experiencing ephemeral grounding.

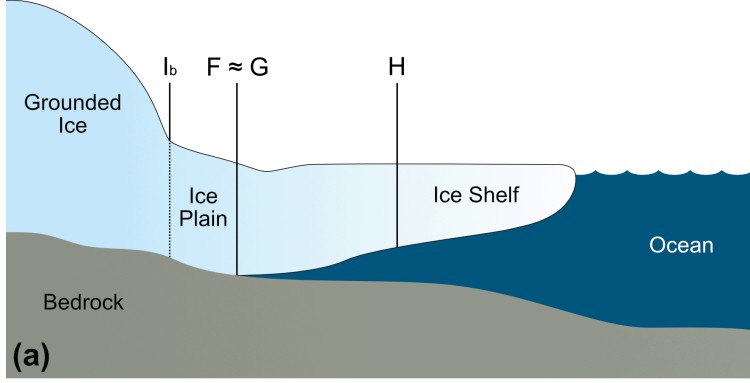

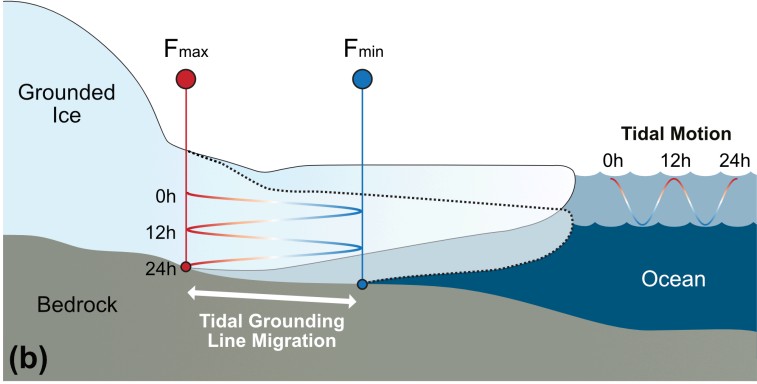


**Figure 1:** (a) Idealised cross section of an ice plain GZ, showing the location of typical satellite-derived GZ proxies: Point $I_b$, the first break-in-slope; Point F, the inland limit of tidal flexure (represented here as approximately equal to Point G, the true GL); Point H, the inshore limit of hydrostatic equilibrium. (b) Idealised cross section of an ice plain GZ experiencing tidal GL migration, showing the difference in position of the ice shelf and migration of Point F between high tide ($F_{max}$) and low tide ($F_{min}$), across 24 hours of a simplified
~12 h semidiurnal tide cycle.



## 2.2 Methods for detecting the grounding zone

GZ surface proxies can be measured *in situ* using instruments such as tiltmeters and GNSS (Stephenson et al., 1979; Smith,
1991; Vaughan, 1995), or using satellite techniques to obtain more widespread measurements (Rignot, 1996; Friedl et al.,
2020). Static satellite methods identify the break-in-slope, Point $I_b$, either from shadows in a single-epoch optical satellite
image (Scambos et al., 2007; Bindschadler et al., 2011) or single-epoch elevation profiles from radar and laser satellite
altimetry (Hogg et al., 2018; Brunt et al., 2010a; Li et al., 2022a). Dynamic satellite methods use data acquired at two or
more distinct phases of the tidal cycle to locate the limits of tidal flexure, Points F and H. CryoSat-2 radar altimetry has been
used to map Point F by applying a pseudo-crossover method to detect tidal flexure (Dawson and Bamber, 2017, 2020). This
provides good spatial coverage by assimilating large data volumes over a multi-year record, but cannot be used to detect
short-term GL migration between specific tides. Differential Interferometric Synthetic Aperture Radar (DInSAR) is one of
the most commonly used dynamic satellite methods, and can produce quasi-continuous maps of Points F and H from double-
differenced interferograms produced from three or four SAR images acquired at different times in the tidal cycle (Rignot,
1996). However, the temporal sampling and spatial coverage of DInSAR in Antarctica is limited, in part due to its reliance
on a short repeat-time between acquisitions to retain interferometric coherence. Recent studies using data from the COSMO-
SkyMed (CSK) constellation, with a short repeat-period of 1, 3, 4 or 8 days and up to 3 m spatial resolution, have improved
DInSAR sampling of tidal GL migration in specific locations (Milillo et al., 2017; Minchew et al., 2017), but are still
complicated by the fact that interferograms are produced from oblique imagery from at least three different tide times.
Moreover, while the relatively short-repeat period of CSK enables interferometric coherence to be maintained on faster
floating ice streams, the spatial coverage of coherent SAR image pairs is still limited, constraining the period where
continent-wide studies can be performed.

An alternative dynamic satellite technique is repeat-track satellite laser altimetry (RTLA), which was first introduced for GZ
detection by Fricker and Padman (2006). For this method, comparison of repeat tracks of ice shelf elevation profiles sampled
at different tidal states identifies elevation anomalies that relate to Points F and H along discrete ground tracks. RTLA was
pioneered using data from the Geoscience Laser Altimeter System instrument on board the Ice, Cloud and Land Elevation
Satellite (ICESat) that was in orbit from 2003 to 2009. However, ICESat only had a single ground track that only repeated to
about ±100 m, which led to unrecoverable topographic biases across GZs (Fricker et al., 2009). In contrast, the Advanced
Topographic Laser Altimeter System (ATLAS) that launched on board ICESat-2 in 2018 has a six-beam design with more
accurate pointing, which reduces across-track deviation from the reference ground track (RGT), providing better spatial
sampling of the GZ. Only two repeat measurements are required to locate Point F using RTLA, providing an advantage over
DInSAR which requires at least three repeats, and allows us to attribute the migration of Point F to individual tidal states. Li
et al. (2022c) has used these data to locate a single Point F, H and $I_b$ (Fig. 1a) along most ICESat-2 ground tracks around





Antarctica. Here, we extend the Li et al. (2022c) record at Bungenstockrücken to locate *multiple* Point F positions along each
ICESat-2 ground track as it migrates over the tide cycle (Fig. 1b), providing novel observations of tidal GZ behaviour.

## 3 Data and Methods

### 3.1 Datasets

The ATLAS instrument onboard ICESat-2 is the first spaceborne photon-counting laser altimeter (Markus et al., 2017). It
samples 11 m diameter footprints (Magruder et al., 2021) estimating surface heights every 0.7 m along its 1387 RGTs.
ATLAS transmits a single beam of 532 nm (green) laser light pulsing at 10 kHz along each RGT, which is split into six
beams organised into three pairs, each separated by 3.3 km. The beams in each pair, one 'weak' and one 'strong', are
separated by 90 m across-track, and each beam follows its own ground track (*GT1L, GT1R, GT2R...to GT3L*), although the
mapping of the weak/strong beams to each ground track per pair switches roughly twice per year as ICESat-2 switches
orientation to maximise solar illumination of solar panels. The system operates in a 91-day exact repeat orbit, providing
repeat surface height measurements along each ground track about four times per year (Neumann et al., 2019). We obtained
data from version 5 of the Land Ice Height (ATL06) product along all ground tracks crossing the Bungenstockrücken GL
(Fig. 2b). This included data from repeat cycles 3 to 15 (April 2019 to May 2022; Smith et al., 2021), with cycles 1 and 2
omitted due to instrument off-pointing at the start of the mission. ATL06 provides surface height measurements every 20 m
along-track, calculated by averaging the L2 geolocated photon data over 40 m segments (Smith et al., 2019). We observed
no significant differences between 'weak' and 'strong' beams in the ATL06 product, therefore we included all beams in our
analysis.

We obtained coincident tide amplitudes at the most seaward point of each ICESat-2 ground track per cycle from the circum-
Antarctic inverse tide model CATS2008 (Howard et al., 2019) using the pyTMD software (Sutterley et al., 2019).
CATS2008 is an update of the model described by (Padman et al., 2002) that is widely considered the best performing tidal
model in the region, in part due to its assimilation of ICESat altimetry data over large ice shelves (King et al., 2011). We
selected three locations 35 – 40 km seaward of Bungenstockrücken (Fig. 2a) to extract a time series of tide heights over a
single year (centred on 01/01/2020), which are used to calculate annual tide distributions across the width of the region.

### 3.2 Repeat-track laser altimetry

RTLA is a dynamic method for GL estimation that uses repeat measurements made at different phases of the tidal cycle to
estimate the temporal change in the surface elevation of the ice shelf caused by ocean tides to identify the limits of tidal
flexure (Points F and H) as proxies for GL location (Fricker and Padman, 2006). This method typically involves three steps:





|  |  |  |
|---|---|---|
| (i) | Data preparation; | |
| (ii) | Define a reference elevation profile; | |
| 165 | (iii) | Locate the limits of tidal flexure by calculating elevation anomalies relative to the reference profile. |

In this study we present updates to steps (ii) and (iii) that allow us to improve the temporal sampling of short-term tidal GL migration. We describe each of these steps below.

### 3.2.1 Data preparation

We obtained ATL06 surface height data along all ICESat-2 ground tracks in the study area (Smith et al., 2021; Scheick and
others, 2019). We removed poor quality measurements caused by cloud cover, blowing snow or background photon clustering using the *ATL06_quality_summary* parameter. We rejected all values either without a valid associated across-track-slope measurement and/or where the minimum segment difference exceeds 1, which is calculated as the minimum absolute difference between each segment's endpoints and those of its two neighbours (Arendt et al., 2020). Following the method described in Li et al. (2020), we applied a cross-track slope correction to minimise the effects of rough terrain (Smith
et al., 2019), and for each ground track omitted repeat cycles with fewer than 50% valid measurements along-track, as they are deemed unreliable for GZ calculation. We then performed RTLA analysis along each ground track using the remaining high quality repeat cycles, as described in Sects. 3.2.2 and 3.2.3.





**Figure 2:** Method used to define three types of reference profile for elevation anomaly calculations along ICESat-2 ground tracks, as illustrated for RGT 559 GT3L at the Bungenstockrücken ice plain. (a) Location of Bungenstockrücken on the Ronne Ice Shelf. Red diamonds show the locations used to calculate annual tide distributions. (b) ICESat-2 ground tracks crossing the MEaSUREs and ASAID GLs in the study area (Rignot et al., 2016; Bindschadler and Choi, 2011). (c-e) Repeat surface elevation profiles and anomalies of RGT 559 GT3L, each using a different reference profile: (c) *mean* profile of cycles 4, 6, 8, 9, 11, 12, 13; (d) *neutral* profile of cycle 13; (e) *lowest-sampled* tide profile of cycle 9. The reference profile at zero-elevation anomaly is shown as a black dashed line, and the latitude of the derived Point F positions for each cycle along-track as coloured circles along the x-axis. Dashed coloured lines show the modelled tide height at the time of each cycle at the most seaward point along-track (in (e) these are differenced from the tide of cycle 9). (f) Histogram of CATS2008 modelled tide heights experienced over a single year, centred on 01/01/2020 (Howard et al., 2019), with vertical coloured dashed lines showing the tide height at each repeat cycle. (g) Tidal time series showing tide height, velocity and phase during the 24 hour period before and after the ICESat-2 overpass for each repeat cycle at RGT 559 GT3L.



### 3.2.2 Defining the reference elevation profile

The reference profile is important for identifying the limits of tidal flexure, as it is used as a baseline from which to calculate elevation anomalies for each ICESat-2 repeat sampled at different tidal states. The *mean* elevation profile has been used as the reference profile in all previous RTLA studies (Fricker and Padman, 2006; Fricker et al., 2009; Brunt et al., 2010b, 2011; Li et al., 2020, 2022a, b), but here we propose two alternative approaches: using a reference profile sampled at a *neutral* tide or at the *lowest-sampled tide*. These are illustrated for RGT 559 GT3L in Figure 2. Before selecting one of these approaches for our study, we first evaluated the results from each and considered the situations for which each would be appropriate. To test each approach, we fitted a cubic B-spline approximation to the valid repeat surface elevation profiles per ground track, using a smoothing parameter of 0.7. The use of an interpolated profile rather than the raw data points is more robust, providing a method for handling small along-track data gaps in individual profiles, which can be consistently applied across all repeat tracks.

(i)   *Mean* **profile** (Fig. 2c): The *mean* elevation profile is calculated as the mean of the interpolated profiles per repeat cycle along each ground track. However, this reference profile does not represent the true "mean" ice shelf surface corresponding to zero ("neutral") tide. Instead, it gives us the along-track mean of the ice shelf surfaces at the tide phases sampled by ICESat-2. This can introduce an observation bias; for example, the *mean* reference elevation profile would be biased high if most sampled repeats of a single RGT are acquired at positive tides, or during one or two extreme high tides. This particularly affects areas experiencing large tidal GL migration, as can be seen clearly in Fig. 2(c), where the anomalies for cycles 6 and 13 fall below zero between -80.96° S and -80.88° S. This might incorrectly suggest that tidal flexure extends this far inland, when in fact it reflects the skew of the mean reference profile caused by GL migration further inland at higher tides (cycles 4 and 12). Given that we locate Point F along-track where the anomaly gradient for each cycle first deviates from 0, this would lead us to incorrectly locate just two Point Fs along this ground track: one for cycle 4 at ~-80.99° S  and one for all other cycles together at ~-81.96° S (Fig. 2c). As a result, we see a systematic inland bias in the derived GL position when using the mean reference profile, leading to a large underestimation of the total extent of tidal GL migration along this ground track.

(ii)  *Neutral* **tide profile** (Fig. 2d): To more closely represent the true "mean" ice shelf surface, we can use the elevation profile from a single cycle sampled at a *neutral* tide as the reference profile. Ideally, this would use the profile sampled close to a 0 m tide during the neap phase, such as cycle 13 on Bungenstockrücken RGT 559 GT3L (Fig. 2g). Using the neutral tide profile removes the effect of the skewed mean in the inner regions of the GZ and improves our interpretation of flexure between tides, as we are now directly comparing the surface profiles between two individual tidal states for each cycle.  However, it still raises the same issue that for all tides below "neutral", we would locate Point F too far inland, where anomalies first deviate from 0 (e.g. cycles 9 and 11 in Fig. 2d). We



are also unable locate Point F for cycles sampled close to the neutral tide as there is no deviation in elevation from the reference profile (cycle 6 and 13 in Fig. 2d). Again, this would mean we would underestimate the total extent of tidal GL migration, by overlooking migration that occurs at tides below neutral. Moreover, many ICESat-2 RGTs have not sampled a nearly "neutral" tide, so it would not be possible to apply this approach consistently across multiple ground tracks.


(iii) ***Lowest-sampled* tide profile** (Fig. 2e): Alternatively, we propose that the elevation profile of the repeat cycle sampled at the *lowest* coincident tide can be used as the reference profile. This also overcomes the issue of the *mean* reference profile being skewed by inland flexure at higher tides, but with the additional advantage that it can be applied consistently across RGTs, enabling automation. Most importantly, it allows us to locate Point F for cycles sampled at lower tides that are missed when using both the *mean* and *neutral* reference profiles. For RGT 559 GT3L we identify five individual Point Fs using the *lowest-sampled* reference profile (Fig. 2e) as opposed to two with the *mean* (Fig. 2c) or four using the *neutral* (Fig. 2d). By locating the most seaward Point F (from cycle 11 in Fig. 2e) it gives us a more realistic baseline for the low tide GL position against which to quantify the extent of migration at each higher sampled tide. Note that we cannot locate Point F for the lowest-sampled tide itself (i.e. cycle 9 in Fig. 2e) as there is no lower profile to compare to.



The choice of reference profile (*mean / neutral / lowest-sampled tide*) to calculate elevation anomalies ultimately depends on the scientific question being posed and the tidal sampling by the ICESat-2 RGTs. We must also consider the impact of signals in the repeat elevation profiles that are not associated with tidal displacement, which can be especially prominent where there are large time gaps between repeat cycles. For example, in regions of fast flowing ice, elevation gradients will advect through the GZ between repeat passes, or alternatively surface mass balance or long-term ice dynamic thickness change could alter the repeat track elevation profiles and obscure the tidal signal. These issues could be addressed by using shorter-sub-sections of the whole time series to define the choice of reference profile.


For this study we chose to apply the *lowest-sampled* tide reference track for RTLA analysis across the whole three-year time series as we found it to be the most effective for assessing *tidal* GL migration. Regardless of the chosen reference profile, the results will be limited by the range of the tidal amplitudes that have been sampled by ICESat-2 along each RGT. The CATS2008 tide model can provide context for this, for example Fig. 2(f) shows that along RGT 559 the lowest 3% and highest 7% of tides have not been sampled, meaning we are unable to locate the full extent of the migration of Point F at these most extreme tides. The tide phase and velocity at each repeat cycle can also affect the observed flexure profile, therefore we also extract this information from CATS2008 to aid our interpretations (Fig. 2g).




### 3.2.3 Locating the limits of flexure in regions of ephemeral grounding

Most previous GZ studies using RTLA located a single Point F per track (Fricker and Padman, 2006; Fricker et al., 2009; Brunt et al., 2010b; Li et al., 2020, 2022a, b). However, with our method using the *lowest-sampled* tide as the reference profile, it is possible to locate multiple Point F positions along each ICESat-2 ground track, each sampled at a different tidal state. To achieve this, we calculated the elevation anomalies per repeat cycle against the reference profile for each ground track, as is shown for RGT 559 GT3L in Fig. 3. The region where the elevation anomaly is close to zero is interpreted as fully grounded ice, and where it is close to the modelled tide prediction with zero gradient as freely floating ice in hydrostatic equilibrium. We compared this with modelled tide predictions from CATS2008 to distinguish the tidal signal from other causes of elevation change, including surface mass balance and ice dynamics. We then located Point F for each repeat cycle using an automated technique adapted from Li et al. (2022). First, we applied a low-pass Butterworth filter to the elevation anomalies per cycle, using a normalised cut-off frequency of 0.016 and an order 5, to remove the high-frequency noise whilst retaining the shape of each anomaly curve. From this, we located Point F where the anomaly gradient first deviates from zero and the 2[nd] derivative of the anomaly peaks (Fig. 3b-g). To ensure the correct estimation of Point F, we restricted the choice of peak to where anomaly values <0.25 m and then manually adjusted any choice of peak where it was still visibly incorrect. For each Point F we then extracted the latitude and longitude as well as the coincident tide height and phase at the time of ICESat-2 overpass from CATS2008, which allows us to map the migration of Point F across the tide cycle (Fig. 3h). Finally, to investigate modes of tidal GL migration along each ground track we plotted the distance between Point F and the location of the furthest seaward Point F against the coincident tide height for each repeat cycle, distinguishing between those sampled during rising vs falling tides (Fig. 3i).

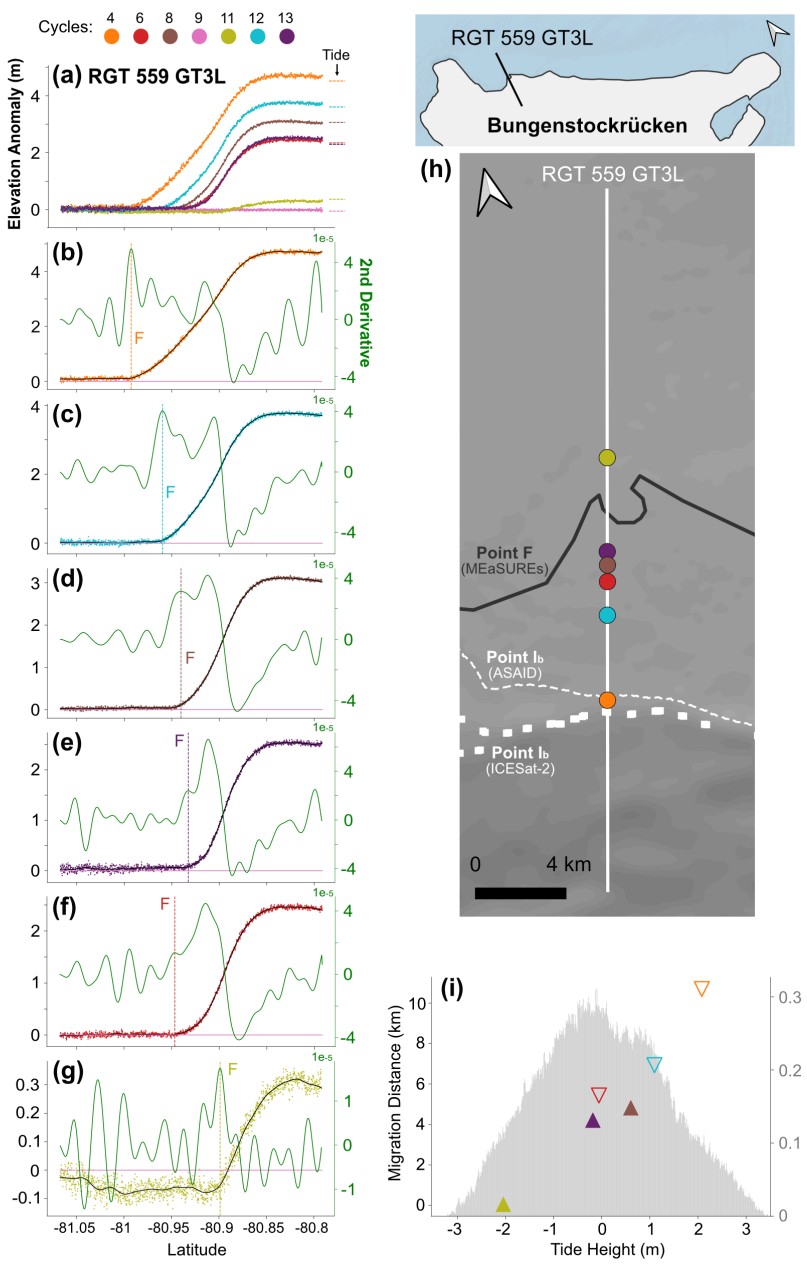

**Figure 3:** Method used to locate Point F per repeat cycle from ICESat-2 elevation anomalies, as illustrated for RGT 559 GT3L. (a) Elevation anomalies for repeat cycles 4, 6, 8, 11, 12 and 13, each calculated against the lowest-sampled tide reference profile of cycle 9. Dashed coloured lines indicate the CATS2008 modelled tide height per cycle, differenced from the cycle 9 tide (Howard et al., 2019). (b-g) Individual elevation anomalies for each repeat cycle, with the low-pass-filtered anomalies in black and the 2nd derivative of the filtered anomalies in green. Vertical dashed lines shows the along-track latitude where Point F has been located from the peak(s) of the second derivative. (g) Map showing the derived position of Point F per repeat cycle along RGT 559 GT3L. (h) Along-track migration distance of Point F per cycle for RGT 559 GT3L as a function of coincident tide height, overlain on the histogram of tide heights experienced over a single year (centred on 01/01/2020).



### 3.3 Study region

We tested our updated RTLA methodology along all ICESat-2 ground tracks crossing a 220km section of GL at the ice plain
north of Bungenstockrücken, located on the southern Ronne Ice Shelf in between the Institute and Möller ice streams (Fig.
2). This is an ideal location to study tidal GL migration for four reasons:

(1)   **Large tidal range with ephemeral grounding**: The southern Ronne experiences predominantly semidiurnal tides
with a 7-8 m total tidal range, one of the largest in Antarctica (Padman et al., 2018). Ephemeral grounding has already
been identified over this ice plain in earlier ICESat studies (Brunt et al., 2011);

(2)   **ICESat-2 sampling**: The high latitude location, GZ orientation and low cloud cover leads to a high density of
unobstructed ICESat-2 tracks crossing normal to the GL (67 RGTs, each with six separate ground tracks);

(3)   **Low ice velocity**: Ice flows across the GL at ~5 m a$^{-1}$ (Rignot et al., 2017), resulting in little advection of surface
features between repeat profiles over the three year study period, which could otherwise affect the success of RTLA;

(4)   **Evidence for past change**: There is evidence that this region has undergone significant GL retreat in the past, and as
the upstream ice sheet is currently grounded over a >1.5 km deep basin with a retrograde slope it is potentially
susceptible to future ice loss (Ross et al., 2012; Siegert et al., 2013). In general, ice plains are a good location to
monitor early signs of change as they are highly sensitive to small perturbations in ice dynamics, sea level, surface
mass balance, basal melt and sediment deposition (Brunt et al., 2011; Horgan et al., 2013b).

We use five existing GL datasets at Bungenstockrücken to compare to our results, as summarised in Table 1.

**Table 1:** Existing grounding line datasets used in this study to compare to our ICESat-2 RTLA results.

| Grounding Line Dataset | Continuous Line / Discrete Points | Satellite(s) | Method | Acquisition Date | Reference |
|---|---|---|---|---|---|
| **MEaSUREs Point F** | Line | RADARSAT-2 | DInSAR | 2009 | Rignot et al. (2016) |
| **ICESat Point F** | Points | ICESat | RTLA | 2003-2009 | Brunt et al., (2010a) |
| **ICESat Point "F2"** (The maximum inland flexure limit at high tide along eight individual ICESat ground tracks at Bungenstockrücken) | Points | ICESat | RTLA | 2003-2009 | Brunt et al. (2011) |
| **ICESat-2 Point F** | Points | ICESat-2 | RTLA | March 2019 – September 2020 | Li et al. (2022c) |
| **ASAID Point I$_b$** (Antarctic Surface Accumulation and Ice Discharge Project) | Line | Landsat-7 ICESat | Photoclinometry and laser altimetry | Landsat-7: 1999-2003 ICESat: 2003-2009 | Bindschadler and Choi (2011) |
| **ICESat-2 Point I$_b$** | Points | ICESat-2 | Laser altimetry | March 2019 – September 2020 | Li et al. (2022c) |





## 4 Results

### 4.1 Extent of tidal grounding line migration at Bungenstockrücken

Our results reveal the spatial pattern of tidal GL migration at Bungenstockrücken across a ~1,300 km$^2$ zone of ephemeral grounding (Fig. 4). Across the central part of the study area (between Areas A and B), we observe the inland limit of tide
flexure (Point F) to migrate by 5 to 15 km across the GZ, although the total migration distance may be greater in regions where the full tide range has not yet been sampled by ICESat-2. In this central region, Point F measured at the highest tides consistently reaches several kilometres further inland than the MEaSUREs GL, and as far inland as the ICESat-2 break-in-slope.  The spatial pattern of short-term tidal GL migration is not consistent across the region, with the most migration observed in Areas A and B (Fig. 4a).

In Area A and across the central section we observe a fairly consistent migration of Point F with the tide. This is exemplified by the pattern of RGT 559 GT3L (Fig. 4b,c), where we observe Point F migrating progressively further inland as tides increase, totalling 10.4 km of GL migration across the ~4.5 m sampled tide range (90.3% of total tide range). At the highest sampled tide of +2.08 m (cycle 4), Point F reaches as far inland as Point I$_b$, but since 7% of tides experienced in the region exceed +2.08 m, it is possible that flexure extends even further inland at these higher tides. In Area B, we observe two main
clusters of Point F, possibly indicating the presence of two stable GL locations (one landward and one seaward) depending on the tide. This pattern can be seen in the surface and anomaly profiles of RGT 574 GT3L (Fig. 4d,e), which show a ~11.5 km separation between the seaward GL position near -81.45° S and the landward GL position near -81.55° S. Similarly to RGT 559 GT3L, the landward GL position at the highest sampled tide for RGT 574 GT3L coincides with the ICESat-2 Point I$_b$, just inland of the large surface undulation visible in the elevation profile (Fig. 4d). We note that in parts of Area B the
landward GL position extends up to 5 km further inland than the ASAID Point I$_b$, up to 3 km beyond the ICESat-2 Point I$_b$, and is positioned around the prominent surface undulations visible in the ice surface topography.

In the areas west of Area A and east of Area B, Point F is consistently located just inland (<2 km) of the ASAID and ICESat-2 break-in-slope with little tidal migration. This is consistent with the surface features of a typical GZ (i.e. non-ice plain) (Schoof, 2011), indicating that these regions are beyond the edges of the ice plain.





**Figure 4:** Tidal GL migration at Bungenstockrücken, measured by ICESat-2 RTLA. (a) The location of Point F at every sampled repeat cycle per ICESat-2 ground track is coloured by % of maximum tide height, overlain on the REMA 8m DEM (Howat et al., 2018), MEaSUREs GL (Rignot et al., 2016), ASAID break-in-slope GL (Bindschadler and Choi, 2011), and the most recent ICESat-2 derived break-in-slope (Point $I_b$) along each ground track (Li et al., 2022c). The two areas with the largest observed GL migration are highlighted as Areas A and B, and the locations for RGT 559 GT3L, 537 GT3L and 574 GT3L are shown. (b-c) The repeat elevation profiles and anomalies of RGT 559 GT3L, with repeat cycles coloured by tide (as % of maximum tide height) and labelled in (c). The location of ICESat-2 Point $I_b$ is marked with a vertical dashed line (Li et al., 2022b). (d-e) The equivalent repeat elevation profiles and anomalies for RGT 574 GT3L.

## 4.2 Long-term grounding zone change at Bungenstockrücken

To assess if there has been any long-term GZ change in our study region, we compared our results to previous GL datasets (Fig. 5). By locating multiple Point Fs along each ground track between high and low tide, we have been able to assess almost the entire width of the zone of ephemeral grounding at Bungenstockrücken for the first time. It is difficult to make a direct comparison to other GL products (Fig. 5a,b) which only locate a single Point F position (Rignot et al., 2016; Brunt et al., 2010a; Li et al., 2022c). Nonetheless, we see that across the central section of Bungenstockrücken, the ICESat-2 derived





Point Fs for the highest ~60% of tides are located consistently inland of the MEaSUREs GL, by ~2-10 km (Fig. 5a). The

ICESat derived Point Fs are mostly located towards the middle of the band of our ICESat-2 derived Point Fs (Fig. 5a),

although the maximum inland Point "F2s" (Brunt et al., 2011) match our ICESat-2 derived Point Fs at high tide more

closely. There has been little long-term change in the position of Point $I_b$ between ICESat/ASAID and ICESat-2, apart from a

few locations in Areas A and B where we observe several kilometres of landward migration (Fig. 5c).

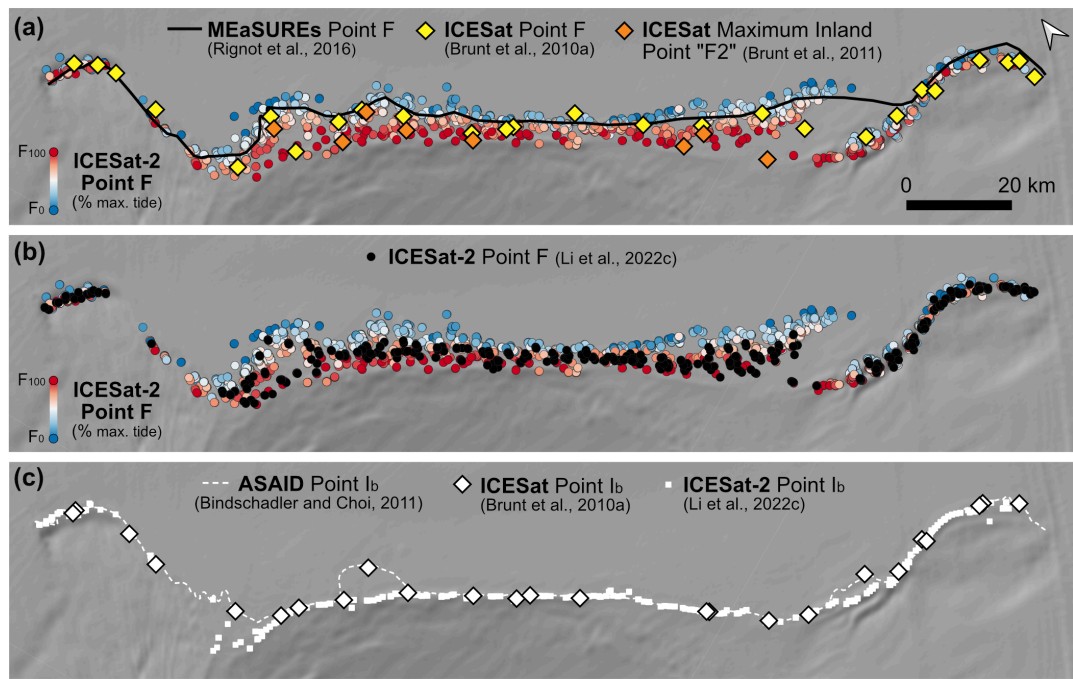


**Figure 5:** (a) Comparison of the ICESat-2 derived Point Fs at Bungenstockrücken to: (i) the ICESat-derived Point F (Brunt et al., 2010a),
(ii) the ICESat-derived maximum inland Point "F2" (Brunt et al., 2011), and (iii) the MEaSUREs GL (Rignot et al., 2016), derived from
2009 RADARSAT-2 data. (b) Comparison of the ICESat-2 derived Point Fs per repeat cycle from this study, to the ICESat-2 derived Point
F dataset from Li et al. (2022c) that locates a single Point F per ground track. (c) Comparison between the Point $I_b$ from ASAID
345 (Bindschadler and Choi (2011); derived from Landsat-7 and ICESat), ICESat (Brunt et al., 2010a) and ICESat-2 (Li et al., 2022c).

**4.3 Modes of tidal grounding line migration**

We investigated the relationship between modelled tide height, tide phase and GL migration distance for the ICESat-2

ground tracks at Bungenstockrücken, and identified four distinct modes of tidal GL migration (Fig. 6). We manually

classified these modes based on visual assessment of the most dominant behaviour. To increase confidence in the

350 classification, we included only tracks where > 2 m of the tide range was sampled and with a total tidal GL migration

distance > 1 km. This removes tracks where small uncertainties in GL position associated with tide phase, basal properties

and long-term surface change are sufficient to obscure the detail in the patterns of GL migration required for this type of

classification.





**Figure 6:** Four modes of tidal grounding line migration observed along ICESat-2 ground tracks at Bungenstockrücken. Three examples are shown per mode, which are classified from top to bottom as: (a-c) Linear, with a linear relationship between tide height and GL migration distance; (d-f) Asymmetric, with higher tides causing an increased rate of GL migration; (g-i) Threshold, with a sharp tide threshold above which there is significant GL migration; (j-l) Hysteresis, with a difference in the impact of tidal variability on GL migration during the rising vs falling tide phase. In each example we show the along-track migration distance between each Point F and the Point F measured at the lowest-sampled tide as a function of coincident tide height from CATS2008 (Howard et al., 2019). Note the variable y axis scale between ground tracks. The orientation of the triangle indicates whether the repeat cycle was sampled during a rising or falling tide phase. Histograms show the distribution of tide heights experienced across a single year, centred on 01/01/2020. (m) Map of Bungenstockrücken showing ICESat-2 ground tracks coloured by migration mode (excluding tracks with <1 km tidal migration or <2 m tide range sampled).



(1)  **Linear** (Fig. 6a-c): In this mode we observe a *linear* relationship between tide height and GL migration distance. Within the sampled tide range, the GL migrates by 2.2 km with every metre of tidal change (a-b), and by 1.7 km/m in (c). Considering a simplified semidiurnal spring tide regime where the tide rises from minimum (-3 m) to maximum (+3 m) within 6 hours, this could translate to an average GL retreat rate of about 2 km/h. We observe little difference between GL positions sampled at rising vs falling tides. However, we note that, for RGT 781 GT1R (b) there is only one F point measured during falling tide, and for all ground tracks we cannot determine whether the linear response extends out to the most extreme tides. The linear mode of tidal GL migration is seen on ground tracks across the study area, but particularly clustered east of Area A (Fig. 6m).

(2)  **Asymmetric** (Fig. 6d-f): In this mode there is an *asymmetric* relationship between tide height and GL migration, with a large increase in the GL migration rate observed at higher tides. For tides below about +0.5 m in (d) and below 0 m in (e) and (f), Point Fs are clustered close together, indicating that there is little tidal GL migration within the lower tidal range. However, as the tide increases beyond this point the rates of GL migration increases sharply to linear rates of +2.5 km/m in (d), +1.8 km/m in (e) and +4.1 km/m in (f). Again, outside of the sampled tide range the magnitudes and modes of GL migration are unknown. Ground tracks with asymmetric tidal GL migration mode are mainly clustered in Area A and just west of Area B (Fig. 6m).

(3)  **Threshold** (Fig. 6g-i): In this mode there is little GL migration up to a certain tide *threshold*, beyond which the GL migrates significantly with just a small increase in tide. The tide threshold in (g) is at ~+0.7 m, and at ~+0.4 m in (h), and in both examples a <0.3 m rise of the tide across the threshold causes the GL to migrate inland by over 10km. At peak spring tide (across the full 6m tide range) this could feasibly occur within 18 minutes, translating to a possible maximum GL migration rate exceeding 30 km/h. Either side of the threshold in (g) and (i), further tide change does not cause much more GL migration. This supports the interpretation that in some locations two fairly stable GL positions (landward and seaward) can exist, as observed in Area B (Fig. 4). Considering the histogram of tide distributions, we can infer from this that the GL is at the landward position ~26% of the time in (g), ~38% in (h) and ~22% in (i). The majority of ground tracks with threshold tidal GL migration mode are located in Area B (Fig. 6m).

(4)  **Hysteresis** (Fig. 6j-l): In this mode there is a *hysteresis* between tidal forcing and GL migration, related to the *phase* of the sampled tide at each measurement of Point F. This means we observe a different GL migration behaviour between rising and falling tides. For example, along RGT 34 GT1R (j) we note a landward and seaward cluster of Point F positions separated by about 15 km, similar to the threshold pattern seen in (g-i). However, instead of a single tidal threshold that defines if the GL is in the landward or seaward position, in (j) we observe two different thresholds for the rising and falling tide phases. From this we infer that at the lowest tides the GL is located in the seaward position, but as the tide rises and crosses some threshold between +0.5 m and +1.2 m there is enough buoyant uplift





force to unground the ice. This then causes ocean water to intrude into the GZ cavity and, due to the likely very flat bed, allows the GL to migrate by up to 15 km to its landward position. As the tide rises beyond +1.2 m, the GL does not migrate any further within the sampled tide range. However, as the tide turns and falls below +0.5 m there is not an immediate re-advance of the GL to its seaward position. Instead, the GL is observed to remain at its landward
position until the tide has fallen beneath some secondary threshold at about -0.8 m. A similar, if less pronounced, hysteresis mode is observed in (k) and (l). Hysteresis is observed on ground tracks across the study area, but particularly in Areas A and B and close to ground tracks with asymmetric and threshold modes (Fig. 6m).

The spatial pattern of tidal GL migration at Area B confirms the presence of a landward and seaward GL, with the location at any given epoch varying between these depending on the tide (Fig. 7a-c). The repeat elevation profiles and anomalies of
RGT 537 GT3L reveal more detail about the hysteresis mechanism that is observed in this region (Fig. 7d-g). Along this ground track, cycles 9 and 12 are sampled at a very similar absolute tide height (~0.25 m higher at cycle 9 vs 12), but their elevation anomaly profiles look very different (Fig. 7f-g). This can be explained by comparing the coincident tide phases, which shows that cycle 9 was sampled during a rising tide, whereas cycle 12 was sampled during a falling tide. The extra bump further inland in the anomalies for cycle 12 (between -81.5° S and -81.57° S; Fig. 7g) indicates that the ice shelf
surface between the seaward and landward GLs has not yet fully recovered from its high tide position. We discuss the potential causes and implications of this in Section 5.

We note that the four tidal GL migration modes described here are not necessarily discrete categories. This is highlighted in Fig. 6(h), which shows that along RGT 1016 GT3R we observe both linear and threshold behaviours in different parts of the tidal cycle. Hysteresis may also be superimposed on top of the other modes, as is arguably visible in Fig. 6(a), but will
become much clearer to identify as more repeats are acquired by ICESat-2. Uncertainty is also introduced where there are gaps in the sampling of certain tidal ranges, for example above +0.5 m in Fig. 6(h, k), and there is subjectivity introduced by the manual classification of modes. This could explain the overlap between tracks with different modes in Fig. 6(m), and particularly those with hysteresis. Our ability to understand and identify these different tidal behaviours will greatly improve as ICESat-2 collects an increasing volume of repeat measurements over the GZ, which should sample more of the tide range.

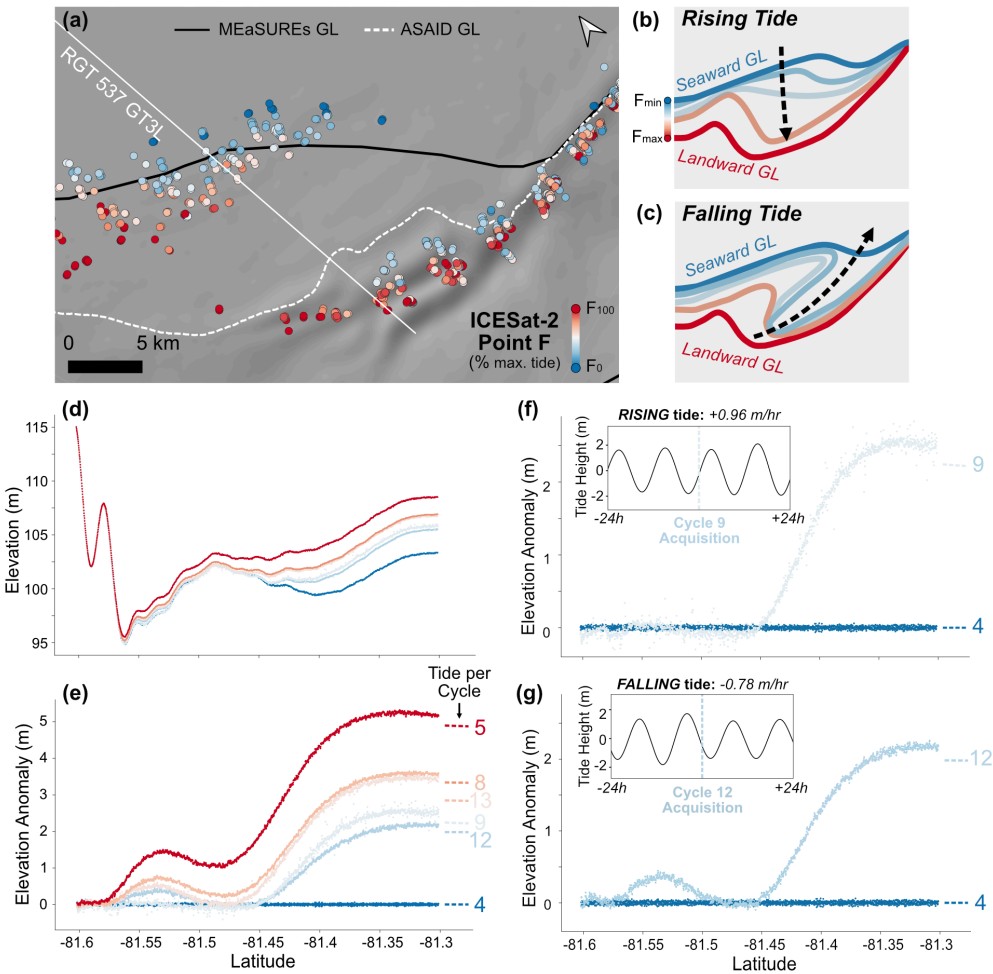


**Figure 7:** (a) The spatial pattern of tidal GL migration observed in Area B, Bungenstockrücken. Point F identified for every repeat cycle of each ICESat-2 ground track is coloured by % of maximum tide height. (b-c) Schematics depicting the flow of water in between the seaward and landward GL positions inferred from ICESat-2 RTLA results in Area B, during the rising and falling tide phase. (d-e) The repeat elevation profiles and anomalies of RGT 537 GT3L, calculated using the *lowest-sampled tide* reference profile from cycle 4. Repeat
cycles are coloured by tide (as % of maximum tide height), and the cycle numbers are labelled in (e) alongside the CATS2008 modelled tide heights, differenced from the tide at cycle 4. (f-g) The individual elevation anomalies for cycles 9 and 12 against the lowest-sampled tide profile of cycle 4. Inset panels show the time-series of tide heights for the 24 hours before and after the time of ICESat-2 overpass at each cycle, showing the coincident tide *phase* and *velocity.*

## 5 Discussion

The improved repeat-track sampling capabilities of ICESat-2 together with our updated RTLA methodology offers several

advantages over other satellite-based techniques for monitoring tidal GZ behaviour. Using this approach, our observation of

widespread short-term tidal GL migration at Bungenstockrücken has implications for how long-term change is assessed and



interpreted, and for how the GL is represented in models. Furthermore, monitoring the dynamic movement of the GL provides an independent satellite method to illuminate bed characteristics and processes that can influence stability in the GZ
region. We expand on each of these implications below.

## 5.1 Advantages of our ICESat-2 RTLA method for monitoring tidal grounding line behaviour

First, the use of ICESat-2 itself offers several advantages that improve the spatial sampling of tidal GL behaviour compared to other satellites. Its track spacing and six-beam pattern greatly increases the density of measurements compared to ICESat, with 402 separate ground tracks crossing the Bungenstockrücken GZ (using the six beams from 67 ICESat-2 RGTs),
compared to 23 ICESat tracks. There is also an improvement in along-track sampling, with ATL06 surface height measurements calculated every 20m along-track (averaged from 40m overlapping segments), compared to a spacing of 172m with ICESat (Schutz et al., 2005). This allows us to better resolve finer-scale details of the shape and configuration of the GL and observe how these may be affected by the tide. Furthermore, its repeat-track configuration and sensitivity to small amplitude tidal variations allow us to attribute GL migration to specific tides, which is not possible with CryoSat-2 radar
altimetry (Dawson and Bamber, 2017). ICESat-2 coverage also extends to 88° S, which is further south than most spaceborne SAR, meaning there is no available coincident DInSAR coverage at Bungenstockrücken during the ICESat-2 period.

Second, the use of our updated RTLA technique improves the temporal sampling of tidal GL behaviour. Most previous RTLA approaches with both ICESat and ICESat-2 use a mean reference profile to derive a single Point F per track (Fricker and Padman, 2006; Fricker et al., 2009; Brunt et al., 2010b; Li et al., 2020, 2022a, b). Our method, by using the lowest-sampled tide as the reference profile, allows us to locate multiple positions of Point F along each ground track as it migrates with the tide. This produces GL measurements at fine temporal resolution, which enables a closer investigation of tidal GZ processes that occur over hourly to daily timescales. Within the time period of this study, we locate up to 11 Point Fs per ground track, each sampled at a different tide, totalling 2402 measurements across the 402 ground tracks. The number of
valid measurements per ground track (of a maximum possible 13) largely depends on the number of repeats obscured by cloud cover. As more cycles of ICESat-2 data are acquired in the future, the temporal sampling of tidal GZ behaviour will continue to increase.

## 5.2 Implications for grounding line monitoring

Our observation of up to 15 km of tidal GL migration across the Bungenstockrücken ice plain is the largest signal reported
anywhere in Antarctica. The signal is an order of magnitude higher than the fastest annual average GL retreat of 1.8 km yr$^{-1}$ observed at Pine Island Glacier between 1992 and 2011 (Park et al., 2013) and several orders of magnitude higher than average long-term Antarctic GL migration rates (Konrad et al., 2018). For comparison, the highest estimated deglaciation





retreat rates from palaeo-records are >2.1 km yr$^{-1}$ and >10 km yr$^{-1}$ for Thwaites and Larsen A, respectively (Dowdeswell et al., 2020; Graham et al., 2022). Therefore, to accurately measure long-term GZ change it is necessary to isolate the spatially

variable pattern of tidal GL migration. At Bungenstockrücken, the observed difference between the MEaSUREs GL and the ICESat-2 derived Point F values (Fig. 5a) could either suggest that there has been up to 10 km of GL retreat since 2009, or no change, depending on whether the MEaSUREs GL was produced from data sampled at high or low tides. This shows the value of providing a precise time stamp in addition to the acquisition date when producing a GL data product, as the tide amplitude at the time of measurement will inform our interpretation about whether a short-term, temporary tidal fluctuation

or a longer-term dynamic change has occurred. Similarly, in this particular location, it is also difficult to quantify any long-term GL migration between the ICESat and ICESat-2 records (Fig. 5a) because the signal is dominated by tidal variability and different methods have been used to derive Point F. The fact that the small number of ICESat-derived maximum inland Point "F2s" from Brunt et al. (2011) are broadly consistent with our highest tide ICESat-2 Point Fs, suggests that there has not been significant retreat of the maximum inland flexure limit over the past 10 to 15 years.

Bungenstockrücken provides a reasonable upper bound for rates of tidal GL migration around the ice sheet, as it is located in an ice plain region with low bed slopes and one of the highest tide ranges in Antarctica. Although tidal migration is likely to be much smaller at most Antarctic GLs (depending on tide range and bed properties, including slope and friction coefficient), even at an order of magnitude less may still be sufficient to obscure early signs of long-term retreat in response to ice shelf thinning (Milillo et al., 2017; Li et al., 2022b) and to impact ice dynamics. Therefore, when assessing long-term

GL migration rates from sparse observations of GL location through time, we must consider the tidal state associated with each measurement epoch. Ideally, long-term GL location estimates should be made at the same tidal state or tide difference; however, given the scarcity of GL measurements in most locations, this has historically not been feasible. In lieu of this, our updated ICESat-2 RTLA methodology could be applied around the ice sheet margin to provide a constraint on the expected tidal range and mechanisms of GL movement. This will enable better evaluation of the impact of tides on previously-derived

estimates of GL position, and ultimately improve the confidence with which we can assess long-term GL change.

The identification of different modes of tidal GL migration using ICESat-2 RTLA (Fig. 6) could be used to identify regions that are particularly sensitive to ice thickness changes, similar to Schmeltz et al. (2001) who showed that areas of ephemeral grounding can reveal subtle changes in ice thickness. In regions with a "threshold" or "hysteresis" mode of tidal GL migration, an observed decrease in the tide threshold that defines whether the GL is located in the landward or seaward

position could indicate a retreat of the GL and signal dynamic ice thinning. Conversely, if the tide threshold rises it could signal GL advance and ice thickening. We suggest that Area B at Bungenstockrücken (Fig. 7) would be an ideal place to monitor this, with its wide zone of ephemeral grounding (up to 15 km) and numerous ICESat-2 tracks exhibiting either threshold or hysteresis GL migration modes. Monitoring ephemeral grounding in this way could enable us to discriminate between inferred thickness changes due to changing firn-air content, and true ice thickness change (Moholdt et al., 2014).



This is useful for detecting early signs of ice sheet change, particularly as the accuracy with which altimeters can measure long-term ice thickness change at the GL is limited by our lack of knowledge about the flotation state of the ice across the GZ (further complicated by tidal GL migration).

## 5.3 Implications for modelling tidal ice shelf flexure and grounding line migration

The representation of GZ processes in ice sheet models strongly impacts projections of future ice sheet mass balance
(Arthern and Williams, 2017). The 15 km observed tidal GL migration at Bungenstockrücken is much larger than standard model grid spacing at the GL, particularly when mesh refinement occurs (e.g. Durand et al., 2009). Similarly, the time step of an ice sheet model is generally much longer than one day; therefore, prescribing a sub-daily change in GL position is not generally possible. It is beyond the scope of this study to parameterise tidal GL effects in ice sheet models at dissimilar scales, but any "short-term fixed" GL derived from observations must be applied in a consistent manner, whether this is
using the highest, lowest or "mean" tide position. This choice of a consistent GL is complicated by the fact that different satellite-derived GL products use different surface proxies (i.e. Points F vs $I_b$), each of which relate differently to the sharp basal stress boundary, which is the parameter that we generally want to represent in models. Moreover, the extent to which tidal GL migration can blur the basal stress boundary in areas of ephemeral grounding is largely unknown.

Our results at Bungenstockrücken show that the use of ICESat-2 RTLA to map Point F across the tide cycle can improve our
understanding of the relationship between short-term GL migration and the long-term ice sheet surface profile. We observe that Point F is located seaward of Point $I_b$ for most of the tidal cycle, but see that at the highest tides it consistently migrates as far inland as Point $I_b$ (Fig. 4). As the break-in-slope ($I_b$) is one of the key surface signatures of the change in basal shear stress at the GL (Schoof, 2011), these observations imply that from a stress-balance perspective the ice sheet geometry is controlled by the furthest inland position that Point F reaches at the highest tides. Although the ice shelf re-grounds seaward
of the break-in-slope at low tides, the variable stresses over the ice plain area do not appear to substantially affect the surface profile. There are substantial deviations between flexure-derived GLs and those derived from break-in-slope, which may therefore be partly explained by sampling of GL positions at either mean or non-maximum tides. We propose that the information derived from ICESat-2 RTLA, by better constraining the extent and pattern of tidal GL migration, could be used to better inform the choice of GL in models where observations are used.

Our results provide observational validations for different numerical models of tidal ice flexure and GL migration. The existence of different modes of tidal GL migration within a region experiencing similar tidal forcing indicates that there is a spatial heterogeneity in local bed properties. Therefore, in regions like Bungenstockrücken that experience large tidal GL migration, models of tide flexure that do not allow for movement of the GL (Holdsworth, 1969; Schmeltz et al., 2002; Walker et al., 2013) cannot produce realistic representations of the basal stress or GL velocity change. Sayag and Worster
(2013) developed an elastic beam model coupled to an elastically deforming bed that allows the GL to migrate



proportionally with tidal forcing.  We observe a similar linear mode of GL migration within the sampled tide range along numerous ICESat-2 ground tracks at Bungenstockrücken (Fig. 6a-c). In contrast, Tsai and Gudmundsson (2015) modelled the GL as an elastic fracture problem forced by the tidally-induced increase in ocean water pressure, concluding that tidally-induced GL migration in areas with prograde bed slopes is asymmetric, with the GL migrating up to 9 times further inland at

high tide than it migrates seaward at low tide. Along several ICESat-2 ground tracks at Bungenstockrücken we observe a similar kind of asymmetry in the mode of tidal GL migration (Fig. 6d-f).  It is likely that the shape and stiffness of the bed exert strong controls over the mode of GL migration (Sayag and Worster, 2013), but independent bed information is generally not high enough resolution to further inform our interpretation of the flexure patterns observed by our ICESat-2 RTLA method.

Our ICESat-2 observations can also inform the representation of elastic and viscous stresses in these models of tidal flexure and GL migration. Both the Sayag and Worster (2013) and Tsai and Gudmundsson (2015) models use purely elastic frameworks, justified due to the short timescales (<12 h) over which the forcing acts. In reality, ice deforms viscoelastically (Reeh et al., 2003) and so these models overlook any time-dependent viscoelastic deformation that follows the initial elastic response to tides. The hysteresis mode of tidal GL migration observed at Bungenstockrücken (Figs. 5j-l and 6) reveals a time

delay between the response of the ice shelf surface (and perhaps also the subglacial till) to tidal forcing. Similar observations in Greenland have shown a difference in the surface deflection profile between rising and falling tides, and have been well described by viscoelastic beam models that capture this lag between the change in the tide and the ice shelf response (Reeh et al., 2000, 2003; Wild et al., 2017). Currently no models of tidal flexure exist that use both a viscoelastic framework and allow for migration of the GL, but this may be necessary to better represent these kinds of processes in GZs subject to high

tidal variability and could account for the discrepancies between model predictions and observations.

Modes of tidal GL migration and why they differ spatially, can also provide insights into the underlying bed topography. As a specific example, we expect that the threshold mode of GL migration (Fig. 6g-i) is likely to only exist in regions with a very flat bed where the ice is very close to flotation, whereby a small increase in tide across this threshold can provide enough buoyant force to unground the ice and cause large GL migration. Area B at Bungenstockrücken is likely grounded

over a very flat bed, as we observe a cluster of ground tracks with a threshold tidal GL migration mode (Fig. 6m). While Brunt et al. (2011) presented a method to calculate bed slopes from tidal GL migration along ICESat tracks based on a hydrostatic assumption, this assumed a linear relationship between migration distance and tide height, which, as our findings prove, is not necessarily valid in the GZ. Our results therefore support the conclusion of Tsai and Gudmundsson (2015) in urging caution when quantitatively inferring bed slopes from measurements of tidal GL migration.

The distribution of tidal GL migration *modes* as well as the *width* and *rate* of tidal GL migration around Antarctic GZs is valuable to inform the choice of parameterisation of tidal flexure mechanisms, basal shear stress and/or basal melt in larger-



scale ice sheet models. For example, Mosbeux et al. (2022) considered different GL migration parameterisations to assess the impact of seasonal sea surface height variability on ice velocities over the Ross Ice Shelf. The first considered an *asymmetric* treatment of GL migration with elastic fracture mechanics introduced by Tsai and Gudmundsson (2015)) and a

constant bed slope, and the second defined migration using hydrostatic equilibrium of the GL with BedMap2 bed slopes. They found that the asymmetric treatment led to larger modelled ice shelf velocity variability that more closely matched GNSS observations. Given that the mechanism of migration varies spatially, even across relatively small distances (as we have demonstrated), then obtaining reliable observations of the spatial distribution of these GL migration modes around Antarctica is needed to inform the parameterisation of GL processes in larger ice sheet modelling studies. Our study at

Bungenstockrücken has successfully demonstrated how ICESat-2 RTLA can be used as a first step towards this.

**5.4 Insights into tidal processes in the ice shelf-ocean-subglacial system**

The cyclical, twice-daily ungrounding and tidal flushing of ocean water up to 15 km into the GZ has implications for several processes contributing to the net dynamics of this system (Fig. 8). Many of these processes are poorly understood with few observations, and so the use of ICESat-2 RTLA to study tidal GL behaviour in settings like Bungenstockrücken could

provide important new insights and constraints.

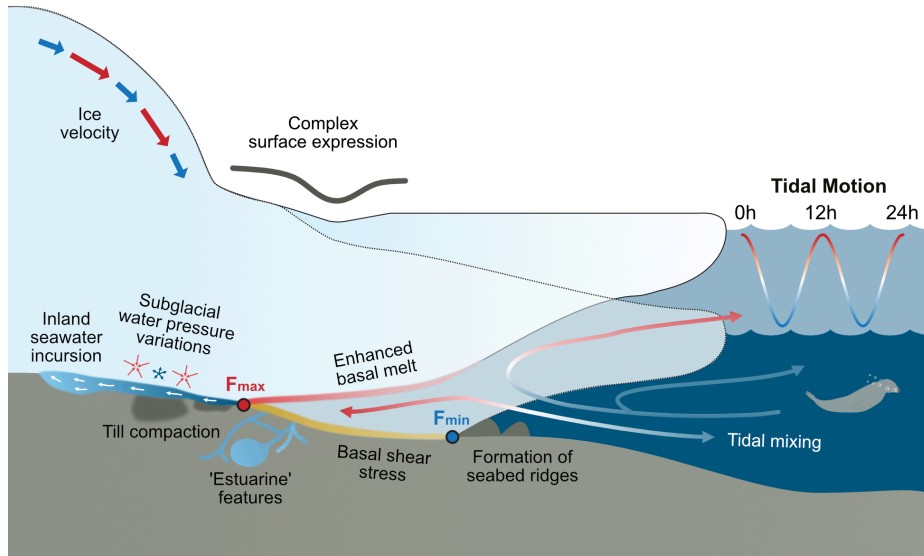

**Figure 8:** Schematic summarising impacts of tidal flexure and GL migration on the ice shelf-ocean-subglacial system. These include impacts on: Ice velocity (Gudmundsson, 2007; Thompson et al., 2014; Robel et al., 2017, 2022; Mosbeux et al., 2022); GZ surface profile (e.g. Schoof, 2011); Inland seawater incursion (Warburton et al., 2020; Robel et al., 2022); Variations in subglacial water pressure (Rosier

et al., 2015; Begeman et al., 2020); Formation of estuarine features in the GZ (Horgan et al., 2013a); Enhanced basal melt (Makinson et al., 2011; Mueller et al., 2012); Basal shear stress (Anandakrishnan et al., 2003); Till compaction, leading to changes in basal lubrication (Walker et al., 2013; Christianson et al., 2013; Sayag and Worster, 2013); Formation of tidal seabed ridges (Dowdeswell et al., 2020; Graham et al., 2022); Ocean circulation through tidal mixing. The landward and seaward tidal GL positions are labelled as $F_{max}$ and $F_{min}$.



Little is known about the oceanography close to most of the Antarctic GZ, but the ability to observe the tidal variability of

cavity geometry with ICESat-2 RTLA will allow us to model the potential melt rates in these anomalous regions. At Bungenstockrücken, our results indicate that an additional ~1,300 km² of ice ungrounds twice-daily between low and high tide (Fig. 4), allowing the relatively warm ocean water to reach deeper into the GZ cavity and exposing this additional area to basal melt. The cold High Salinity Shelf Water (HSSW) beneath the Ronne Ice Shelf causes relatively low basal melt rates (compared to warmer regions like the Amundsen Sea that are affected by Circumpolar Deep Water), but depression of the

seawater freezing point under pressure means it still has the capacity to melt ice at the GL (Nicholls and Østerhus, 2004; Makinson et al., 2011; Naughten et al., 2021). The very shallow water column (<100m) immediately seaward of Bungenstockrücken (Johnson and Smith, 1997) suggests that tidal mixing would be strong here, and is likely to be enhanced as water is "pumped" in and out of the cavity as the GL migrates with the tides. Short-term tidal fluctuations in melt rate and tidal mixing have been shown to impact long-term GL retreat in less stable regions (Graham et al., 2022). Given that

projections of future ice sheet mass loss are highly sensitive to basal melt at the GL (Arthern and Williams, 2017; Goldberg et al., 2019; Adusumilli et al., 2020), the type of information about tidal GZ behaviour that we can derive from ICESat-2 RTLA will be valuable to improve melt parameterisations in ice sheet models.

Variation in basal stresses in the GZ caused by tidal flexure and GL migration can also impact the flow of upstream and downstream ice. Grounding and ungrounding of the ice shelf throughout the tidal cycle has several effects: it changes the

area affected by basal drag; modifies the driving stress through changes in surface slope (e.g. Makinson et al., 2012; Mosbeux et al., 2022); modulates hydrostatic and flexural stresses (e.g. Rosier and Gudmundsson, 2016); and impacts buttressing (e.g. Robel et al., 2017). Rosier and Gudmundsson (2020) concluded that the presence of tides enhances ice flow across the entire Filchner-Ronne Ice Shelf by 21%. Since ice velocities across the Bungenstockrücken GL do not exceed ~5 m a$^{-1}$ (Rignot et al., 2017), the impact of tidal GL migration on absolute flow speed here is likely to be low, but it may be

more significant in faster flowing regions, such as the Rutford Ice Stream (Minchew et al., 2017).

Our results have implications for testing models and hypotheses in subglacial hydrology. It has been suggested that tidal ice shelf flexure acts to "pump" ocean water upstream of the GL into the subglacial hydrological system, with implications for basal lubrication, basal melt and till compaction (Christianson et al., 2013; Horgan et al., 2013a; Sayag and Worster, 2013; Walker et al., 2013; Drews et al., 2017; Robel et al., 2022). Airborne radar data indicate the possible presence of brackish

water several kilometres inland of the Bungenstockrücken GL (Corr, 2021), which may provide evidence for upstream seawater incursion driven by tidal GL migration. Modelling of this process by Warburton et al. (2020) indicated that water is pumped upstream into the subglacial environment much faster during a rising tide than it drains as the tide falls. The hysteresis mode of GL migration observed at Bungenstockrücken (Figs. 6j-l and 7) suggests there is a similar lag in the response of the ice shelf surface to tidal forcing between rising and falling phases, which could provide observational

support for this theory. For example, the pattern observed in Area B (Fig. 7) is likely to indicate that a subglacial channel or





cavity exists in between the seaward and landward GL positions that fills with water at high tide (above a certain threshold). As the tide falls, the water drains away more slowly and the ice around the seaward GL position (a possible topographic high) re-grounds before the landward cavity has fully emptied, forcing the water to be flushed out from this channel to the east (Fig. 7b,c). This could be a similar pattern to tidal beach drainage, and supports the idea of "estuary" type features at the
ice sheet margins (Horgan et al., 2013a). Knowledge of the modes of tidal GL migration around the coast derived from ICESat-2 RTLA will be useful for constraining such regional models.

## 6 Summary and outlook

We have presented an updated method for determining Antarctic ice shelf grounding line (GL) location from ICESat-2 repeat track laser altimetry (RTLA). We show that, by using the elevation profile at the *lowest-sampled* tide as the reference
profile for calculating anomalies, it is possible to locate the inland limit of tidal flexure at different stages of the tidal cycle. This temporal resolution provides insights into grounding zone (GZ) processes on tidal timescales that are difficult to observe with other methods. We have applied the technique to the GZ of an ice plain north of Bungenstockrücken on the southern Ronne Ice Shelf, where we observe >15 km of tidal GL migration. This is the largest reported distance of ephemeral grounding anywhere in Antarctica and demonstrates the necessity of accounting for short-term tidal GL migration
to accurately measure long-term migration. It also calls into question the use of a fixed or slowly moving GL boundary in ice sheet models, which does not represent the full range of GL migration behaviour. Our updated ICESat-2 RTLA technique also allows us to observe different modes of tidal GL migration, and we classify four modes at the Bungenstockrücken ice plain: "linear", "asymmetric", "threshold" and "hysteresis". This can provide observational validation for models of tidal ice shelf flexure, GL migration and subglacial hydrology at the GZ, and offers a tool for future explorations of bed
characteristics, basal melt and basal shear stress in these sensitive parts of the ice sheet. We suggest that monitoring the change in extent and modes of tidal GL migration around Antarctica could provide vital early indications of wider ice sheet change.

Based on the findings of this study, we make four recommendations for future work:

(1)    Any future satellite-derived measurement of GL position should be accompanied with a timestamp (at least to the
closest hour), ideally also with coincident tide height and phase. For DInSAR this information should be provided per image, and for RTLA for both the reference and repeat cycle tides. This will allow a more robust assessment of the impact of tidal processes on individual measurements of the GL position.

(2)    Future studies should scale this analysis up across Antarctica to derive a continent-wide dataset of the extent and mode of tidal GL migration. This could be used to improve confidence in long-term records of GL migration.



(3)  Targeted ground-based surveys in regions subject to extreme tidal variability would be valuable to better understand the link between these surface observations and the processes taking place within and beneath the ice in GZs on tidal timescales. We suggest that Bungenstockrücken would be a suitable location, due to its high tidal signal, low ice velocity and crevassing, and current long-term stability which isolates the tidal signal from long-term change.

(4)  The observations presented here have only been possible to obtain using ICESat-2. Therefore, it is crucial that we

extend our satellite-based *repeat-track* sampling of ice sheet surface elevation beyond this satellite mission to continue this valuable record as ice sheets continue to respond to the changing climate.

*Code Availability.* The code developed for this study can be provided by the corresponding authors upon request and will be made available at a DOI tbc.

*Data Availability.* The ICESat-2 ATL06 data, ICESat-2 grounding zone products and the MEaSUREs grounding line product used in this study are available from the National Snow and Ice Data Center (NSIDC). The ASAID grounding line and ICESat-derived grounding zone products are available from the U.S. Antarctic Program Data Center.

*Author Contribution.* BF and OM planned the research; BF conducted the data analysis and wrote the manuscript. All authors provided insights in the interpretation of data, and reviewed and edited the manuscript.

*Competing interests.* The authors declare that they have no conflict of interest.

*Acknowledgements.* This work was led by BF at British Antarctic Survey (BAS) and the School of Earth and Environment at the University of Leeds. BF was supported by the Natural Environment Research Council (NERC) SENSE Centre for Doctoral Training (NE/T00939X/1). AEH was supported by the NERC DeCAdeS project (NE/T012757/1) and the ESA Polar+ Ice Shelves project (ESA-IPL-POE-EF-cb-LE-2019-834). HAF was supported by NASA grant 80NSSC20K0977. LP

was supported by the ESR ICESat-2 grant 80NSSC21K0911. The authors gratefully acknowledge the National Aeronautics and Space Administration for acquiring the ICESat-2 satellite data.

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
