# Peer review of "Modes of Antarctic tidal grounding line migration revealed by ICESat-2 laser altimetry"

_The Cryosphere, 2022_

## Referee Comment (RC3)

**Review comments for Bryony I. D. Freer et al.'s "Modes of Antarctic tidal grounding line migration revealed by ICESat-2 laser altimetry"**
Tian Li
19 March 2023

**General Comments**

In this research, Bryony Freer and co-authors mapped the short-term variations in grounding line locations at the Bungenstockrücken Ice Plain using ICESat-2 satellite laser altimetry with a new approach of calculating elevation anomalies. They observed > 15 km tidal GL migration and discovered four different modes of tidal GL migration which are useful in validating numerical model simulations of GL migration and understanding the tidal processes of the ice shelf-ocean-subglacial system. Overall, I find this study interesting, robust and provide new insights into tidal GL migrations, which is important in determining long-term GL changes. The paper is well written and the topic fits well in this journal. I have several moderate concerns detailed below and I hope the authors could address them in the revised version.

**Major Comments**

**Inaccurate statements on mean-profile method:**

In **Section 3.2.2**, the authors claim that the mean elevation profile approach cannot be used to calculate the fine-scale short-term GL migrations with ocean tides compared to the newly proposed "Lowest-Sampled Tide" (LST) approach, this is not technically correct.

First, it does not make sense to map short-term GL migrations using a mean elevation profile of all 7 repeat cycles (in the case of RGT 559 GT3L). The nature of using an average elevation profile of all repeat cycles across the study period has already determined that we can only derive one Point F - which is the most landward ice flexure location experienced by the ice shelf among all the repeat cycles. This is the reason you see the clustered Point F features in Figure 2c. But this does not mean that the mean elevation profile method itself cannot be used to derive fine-scale short-term GZ features, which can be achieved by reducing the number of cycles used in calculating the elevation anomalies using a mean elevation profile.

Second, as the authors already mentioned "*only two repeat measurements are required to locate Point F using RTLA*" in **Line 131**, if you iterate different combinations of **any two** ICESat-2 repeat cycles, calculate the mean elevation of these two cycles and estimate Point F from elevation anomalies based on this mean elevation profile, you will get a very detailed spatial sampling of the short-term Point F locations along one ground track, similar to the results in Figure 2e. For example, using only cycles 9 and 11 along RGT559 GT3L with the mean elevation approach can produce a most seaward Point F location between -80.90 and – 80.85 latitudes.

In fact, the mean elevation approach has several benefits over the LST approach proposed in this study:

1) **Denser spatial sampling**: the number of different repeat cycle combinations available for Point F calculation will be higher than LST, because the latter only compares different repeat cycles to one fixed cycle with the lowest-sampled tide.

2) **Higher temporal resolution**: using the neighboring two repeat cycles can provide Point F with a time resolution of 91 day, this cannot be achieved by LST.

3) **Free from errors in tidal models**: the mean elevation approach doesn't need tidal model to determine a reference profile.

Third, in contrast to the statement in **Line 227-229** "*This also overcomes the issue of the mean reference profile being skewed by inland flexure at higher tides, but with the additional advantage that it can be applied consistently across RGTs, enabling automation*", the automation of the iteration of different repeat-cycle combinations with the mean elevation approach is straightforward to implement, actually the method should be similar to the LST method shown in Figure 3. In addition, using only two cycles can also overcome the issues of skewed ice flexure at high tide.

Therefore, I would like to hear the authors' opinion on this point, and would like to see a comparison between Point F locations calculated from any given two repeat cycles using the mean elevation approach and the results from their LST method.

**Tidal model accuracy:**

The Neutral Tide and the LST methods proposed in this study rely on tidal model predictions from CATS2008. Previous research show that the ice at deep grounding zone at narrow ice shelf embayment may not respond adequately in phase with ocean tides (Li et al., 2023; Reeh et al., 2000), which means that tidal model cannot always provide accurate tidal amplitude predictions. If this is the case, will the proposed approach still be valid for deriving short-term Point F locations and how the inaccurate tidal model predictions will impact your Point F result? Please clarify.

**Specific Comments**

**Line 37**: Please rephrase this sentence, GZ can be wider than 10 km as demonstrated by the results of >15 km migration in this study.

**Line 45-50**: This paragraph lacks mentioning previous research in studying the short-term GL migrations using satellite altimetry and DInSAR, such as:

- Brunt et al. (2011): https://www.cambridge.org/core/journals/journal-of-glaciology/article/analysis-of-ice-plains-of-the-filchnerronne-ice-shelf-antarctica-using-icesat-laser-altimetry/80F41E7FDC8124136AF50615861D5C71
- Milillo et al. (2017): http://doi.wiley.com/10.1002/2017GL074320
- Milillo et al. (2019) https://www.science.org/doi/10.1126/sciadv.aau3433
- Brancato et al. (2020): https://onlinelibrary.wiley.com/doi/10.1029/2019GL086291
- Milillo et al. (2022): https://www.nature.com/articles/s41561-021-00877-z

**Line 52-53:** What are the spatial and temporal resolutions of these historical archives? What is the requirement of spatial-temporal resolution for assessing the tidal GL migration?

**Line 63-65:** "*where short-term GL migration is likely to impact both ice dynamics through rapid variations in basal shear stress, and basal melt rate through changes in cavity geometry enhancing tidal mixing.*" Here needs a reference.

**Line 109-110**: Dawson and Bamber (2020) also mapped Point H in addition to Point F in their study.

**Line 111:** Need to mention why CryoSat-2 is not suitable to detect short-term GL changes – due to its long repeat cycle (369 day) (Dawson and Bamber, 2017).

**Line 116-122**: Mohajerani et al. (2021) have also improved the DInSAR sampling under different tidal phases using Sentinel-1a/b SAR images across the Antarctic Ice Sheet.

**Line 128**: Please mention that the "*unrecoverable topographic biases across GZs*" is caused by the large across-track slope across the GZ.

**Line 128-131**: "*In contrast, the Advanced Topographic Laser Altimeter System (ATLAS) that launched on board ICESat-2 in 2018 has a six-beam design with more accurate pointing, which reduces across-track deviation from the reference ground track (RGT), providing better spatial sampling of the GZ.*" Not accurate and need clarification:

1) The six-beam design reduces the across-track deviation from the **Reference Pair Track (RPT)** inside each beam pair, not the Reference Ground Track (See https://nsidc.org/sites/default/files/icesat2_atl06_atbd_r005.pdf).
2) Please give a number of the ICESat-2 point control accuracy here and how it compares to ICESat. Luthcke et al. (2021) show that the performance of ICESat-2 can achieve 4.4 ± 6.0 m, this represents a very small across-track separation across repeat cycles.

**Line 133:** Here I suggest the authors to add some discussions on the pros and cons of Li et al. (2022b)'s results, and talk about why it is necessary to extend the data record in this study.

**Line 142-144:** I think there is no need to mention the switch between weak and strong beams because this study did not use this feature in GZ calculation.

**Line 149-151**: see my previous comment, can also merge this sentence into Line 142-144.

**Line 152**: "*We obtained coincident tide amplitudes at the most seaward point of each ICESat-2 ground track per cycle*". Please provide an average distance between these ICESat-2 seaward points used in the tidal amplitude calculation and a reference (historic) grounding line. This is important because if the seaward point still locates inside the GZ, then the modelled tidal amplitude may not represent the actual ice shelf elevation change in hydrostatic equilibrium. In addition, the orientation of the ICESat-2 grounds tracks are not always perpendicular with the actual grounding line, this can also introduce biases (Li et al., 2020, 2022a).

**Line 157**: Please clarify why the annual tidal distribution is essential in GZ calculation.

**Line 172**: "*minimum segment difference exceeds 1*", I assume this is 1 meter, please add a unit.

**Line 192-195**: Here the authors merely mention the traditional mean elevation approach in calculating elevation anomalies, then directly propose two new methods without any discussions on the disadvantages of the mean profile method. It is not clear why it is essential to develop two new approaches, why mean elevation approach fails to map short-term GL locations, and how the two new methods can further improve these research gaps. I suggest incorporating some of the information in **Line 201-235** to the beginning of **Section 3.2.2**.

**Line 201-213**: As I have mentioned in the major comment, mean profile approach can provide high spatial-temporal resolution GZ calculation, please consider modify the relevant content accordingly.

**Line 229-230**: The mean elevation profile approach can allow the calculation of Point F from repeat cycles both sampled at low tides, please see my major comments, and rephrase this sentence.

**Line 500-501**: "*The 15 km observed tidal GL migration at Bungenstockrücken is much larger than standard model grid spacing at the GL*", what is the standard model grid size at the GL?

**Line 500-503**: "*Similarly, the time step of an ice sheet model is generally much longer than one day; therefore, prescribing a sub-daily change in GL position is not generally possible.*" Not sure why mentioning sub-daily GL change here, especially the results in this paper cannot achieve the sub-daily resolution either.

**Figures 2 c,d,e**:

I assume the reference elevation profile of each different approach is plotted in the upper panel of subfigures c/d/e, however it is very difficult to discern them from the ICESat-2 repeat profiles. Please change the line symbols of reference elevation profile under different scenarios. Here are some suggestions:

1) consider only include one subfigure to show all the ICESat-2 repeat cycles with a colormap.
2) then add a different subfigure, plot three different reference elevation profiles in distinct colors or symbols and add all ICESat-2 repeat cycles in light grey color with low transparency as a background - this is to make sure that the reference elevation profiles can stand out from ICESat-2 profiles and the readers can clearly distinguish these three different reference profiles.

**Figure 6**: Both panels f and l are RGT 1223 1R, why do the Point F locations along the same ground track behave so differently? Is this a labelling error in the figure?

**References:**

Brancato, V., Rignot, E., Milillo, P., Morlighem, M., Mouginot, J., An, L., Scheuchl, B., Jeong, S., Rizzoli, P., Bueso Bello, J. L. and Prats-Iraola, P.: Grounding Line Retreat of Denman Glacier, East Antarctica, Measured With COSMO-SkyMed Radar Interferometry Data, Geophys. Res. Lett., 47(7), e2019GL086291, doi:10.1029/2019GL086291, 2020.

Brunt, K. M., Fricker, H. A. and Padman, L.: Analysis of ice plains of the Filchner–Ronne Ice Shelf,

Antarctica, using ICESat laser altimetry, J. Glaciol., 57(205), 965–975, doi:10.3189/002214311798043753, 2011.

Dawson, G. J. and Bamber, J. L.: Antarctic Grounding Line Mapping From CryoSat-2 Radar Altimetry, Geophys. Res. Lett., 44(23), 11,886-11,893, doi:10.1002/2017GL075589, 2017.

Dawson, G. J. and Bamber, J. L.: Measuring the location and width of the Antarctic grounding zone using CryoSat-2, Cryosph., 14(6), 2071–2086, doi:10.5194/tc-14-2071-2020, 2020.

Li, T., Dawson, G. J., Chuter, S. J. and Bamber, J. L.: Mapping the grounding zone of Larsen C Ice Shelf, Antarctica, from ICESat-2 laser altimetry, Cryosph., 14(11), 3629–3643, doi:10.5194/tc-14-3629-2020, 2020.

Li, T., Dawson, G. J., Chuter, S. J. and Bamber, J. L.: A high-resolution Antarctic grounding zone product from ICESat-2 laser altimetry, Earth Syst. Sci. Data, 14(2), 535–557, doi:10.5194/essd-14-535-2022, 2022a.

Li, T., Dawson, G. J., Chuter, S. J. and Bamber, J. L.: ICESat-2 L4 Grounding Zone for Antarctic Ice Shelves, Version 1, NASA Natl. Snow Ice Data Cent. Distrib. Act. Arch. Cent., doi:https://doi.org/10.5067/RI67B92708M9, 2022b.

Li, T., Dawson, G. J., Chuter, S. J. and Bamber, J. L.: Grounding line retreat and tide-modulated ocean channels at Moscow University and Totten Glacier ice shelves, East Antarctica, Cryosph., 17(2), 1003–1022, doi:10.5194/TC-17-1003-2023, 2023.

Luthcke, S. B., Thomas, T. C., Pennington, T. A., Rebold, T. W., Nicholas, J. B., Rowlands, D. D., Gardner, A. S. and Bae, S.: ICESat-2 Pointing Calibration and Geolocation Performance, Earth Sp. Sci., 8(3), e2020EA001494, doi:10.1029/2020EA001494, 2021.

Milillo, P., Rignot, E., Mouginot, J., Scheuchl, B., Morlighem, M., Li, X. and Salzer, J. T.: On the Short-term Grounding Zone Dynamics of Pine Island Glacier, West Antarctica, Observed With COSMO-SkyMed Interferometric Data, Geophys. Res. Lett., 44(20), 10,436-10,444, doi:10.1002/2017GL074320, 2017.

Milillo, P., Rignot, E., Rizzoli, P., Scheuchl, B., Mouginot, J., Bueso-Bello, J. and Prats-Iraola, P.: Heterogeneous retreat and ice melt of Thwaites Glacier, West Antarctica, Sci. Adv., 5(1), eaau3433, doi:10.1126/sciadv.aau3433, 2019.

Milillo, P., Rignot, E., Rizzoli, P., Scheuchl, B., Mouginot, J., Bueso-Bello, J. L., Prats-Iraola, P. and Dini, L.: Rapid glacier retreat rates observed in West Antarctica, Nat. Geosci., 15(1), 48–53, doi:10.1038/s41561-021-00877-z, 2022.

Mohajerani, Y., Jeong, S., Scheuchl, B., Velicogna, I., Rignot, E. and Milillo, P.: Automatic delineation of glacier grounding lines in differential interferometric synthetic-aperture radar data using deep learning, Sci. Rep., 11(1), 4992, doi:10.1038/s41598-021-84309-3, 2021.

Reeh, N., Mayer, C., Olesen, O. B., Christensen, E. L. and Thomsen, H. H.: Tidal movement of Nioghalvfjerdsfjorden glacier, northeast Greenland: observations and modelling, Ann. Glaciol., 31, 111–117, doi:10.3189/172756400781820408, 2000.

---

## Author Response (AR1)

Comments from the reviewer are given in black.

Author responses are given in red, and **amendments or additions made to the revised manuscript in bold red.**

**RC1: 'Comment on tc-2022-265', Pietro Milillo**

This article discusses the development and application of a technique using ICESat-2 repeat-track laser altimetry to locate the inland limit of tidal ice shelf flexure and resolve the magnitude and temporal variability of tidal grounding line (GL) migration in Antarctica. The authors apply this technique to an ice plain north of Bungenstockrücken, in a region of the southern Ronne Ice Shelf subject to large ocean tides. They observe a 1,300 km2 area of ephemeral grounding over which the GL migrates by up to 15 km between low and high tide and identify four distinct modes of migration: "linear", "asymmetric", "threshold" and "hysteresis". The short-term movement of the GL dominates any long-term migration signal in this location, and the distribution of GL positions and modes contains information about spatial variability in the ice-bed interface. The authors identify four distinct modes of GL migration: linear, asymmetric, threshold and hysteresis. I was surprised when reading about linear and threshold behaviors they did not mention recent well-known studies confirming these results (i.e. Milillo et al 2022 for linear behavior and Milillo et al 2019 for threshold behavior over the Thwaites Cavity). The authors recommend that these observations can be used to validate models of tidal ice shelf flexure, GL migration, and subglacial hydrology at the grounding zone (GZ). They find a 14 km grounding zone that could be explained with the Stubblefield et al 2021 Model. However, I haven't found any reference to this paper in the manuscript. The study concludes with recommendations for future work, including the need for timestamped measurements of GL position accompanied by tide height and phase, continent-wide analysis of tidal GL migration, and improved representation of GL migration behavior in ice sheet models.

The paper is well written, well organized and provides significant results. I encourage the editor to accept this manuscript after few minor revisions.

Many thanks for the review and positive comments on this work.

I believe the authors could further improve the manuscript by referring the aforementioned literature studies as a further independent confirmation of the validity of their findings.

Milillo, P., Rignot, E., Rizzoli, P., Scheuchl, B., Mouginot, J., Bueso-Bello, J. L., ... & Dini, L. (2022). Rapid glacier retreat rates observed in West Antarctica. Nature Geoscience, 15(1), 48-53.

Milillo, P., Rignot, E., Rizzoli, P., Scheuchl, B., Mouginot, J., Bueso-Bello, J., & Prats-Iraola, P. (2019). Heterogeneous retreat and ice melt of Thwaites Glacier, West Antarctica. Science advances, 5(1), eaau3433.

Stubblefield, A. G., Spiegelman, M., & Creyts, T. T. (2021). Variational formulation of marine ice-sheet and subglacial-lake grounding-line dynamics. Journal of Fluid Mechanics, 919, A23.

Thank you for these suggestions, **these have now been added to the revised manuscript:**

- **Milillo et al. (2022) – Lines 43, 554**
- **Milillo et al. (2019) – Lines 43, 63, 583, 618**
- **Stubblefield et al. (2021) – Lines 54, 640**

Comments from the reviewer are given in black.

Author responses are given in red, and **amendments or additions made to the revised manuscript in bold red.**

**RC2: 'Comment on tc-2022-265', Kasia Warburton**

This article addresses the mapping of tidal grounding-line migration from ICESat-2 repeat-track laser altimetry, highlighting the large regions of transiently grounded ice in an ice plain of the Ronne Ice Shelf. The authors raise excellent points about the need for consistency in grounding line products given the scale of tidal grounding-line migration compared to long term retreat rates. They also identify several different patterns of migration, termed "linear", "asymmetric", "threshold", and "hysteresis" - although I agree fully with the authors that given the potential for hysteresis, this categorisation could be highly affected by the gaps in sampling and will be improved by increasing repeats by ICESat-2 (perhaps this point should be made a little louder).

Many thanks for the review and positive comments on this work.

**We have emphasised this final point in the revised manuscript, lines 338-442.**

This is a well-written and significant paper, and the summary and outlook section in particular provides concrete recommendations that will enhance our understanding of grounding-line processes. I have only two minor comments which the authors might like to consider in their final version:

The authors argue convincingly that the choice of lowest-sampled profile provides the best measurement of grounding-line migration. The mean profile is described as introducing observation bias, but the issue appears to be primarily the same issue as with the neutral tidal profile, that the point F cannot be accurately identified when the elevation anomaly is negative. This point could be made more clearly, perhaps with a note about why this limitation exists (e.g. a naive question might be what happens if the highest-sampled profile is taken as reference instead).

Thank you for this comment, we appreciate that this point requires some further clarification in the revised manuscript, and has been addressed in detail in the response to Reviewer 3. The mean profile (when calculated from more than two cycles, i.e. the 'traditional' method), still has the issue with being skewed further inland if there is significant tidal migration at high tide, which makes the interpretation of Point F more ambiguous. This is the key additional issue compared to the Neutral tide method.

Indeed, Point F cannot be located where the elevation anomaly is negative in either case. To address your final question, Figure R1 illustrates the results of using the highest sampled tide profile as the reference (compared to the lowest sampled tide). If we locate Point F at the point at which elevation anomalies start to deviate from 0, then for all of the cycles we would incorrectly locate Point F at approx. -80.99°S.

**Following these comments (and those of RC3), we have added further clarification of the use of the different reference profiles in Section 3.2.2.**

[Figure]

*Figure R1* – Comparison of elevation anomalies for RGT 559 GT3L, using the highest-sampled-tide (cycle 4; upper panel) and lowest-sampled-tide (cycle 9) as the reference profile for calculating anomalies. Coloured circles show the derived Point F in each case. Dashed horizontal coloured lines indicate the modelled tide heights per cycle.

In l.263 the authors describe that they "then manually adjusted any choice of peak where it was still visibly incorrect". From both a reproducibility and automation point of view, this method should be made more explicit.

**Many thanks for raising this. We have added additional explanation of this method in the revised manuscript, as suggested: lines 279-284.**

Ultimately, it is difficult to create an automated processing chain that can deal with all circumstances, particularly in grounding zones (including increased noise and the impact of crevassing and undulating topography, etc.). This could be addressed by filtering out more tracks with poorer fits, however this would lead to a loss of useful data. Therefore, in this study (working in a relatively small area) it was decided to manually check the results and adjust the choice of peak accordingly.

This was done by inspecting the 2nd derivative of the low pass filter for each cycle against the elevation anomalies (as in Figure 3b-g) and checking that the peak in the 2nd derivative selected to locate Point F was in fact that closest to the point at which the elevation anomalies first deviate from zero. The automated method selected the 'correct' peak in the majority of cases, however by visually checking the results, we could adjust the choice of peak where necessary in order to ensure consistent location of Point F between cycles and tracks.

Making the code available (currently tbc) should also be a priority.

**The code is now available in the following repository:** https://doi.org/10.5281/zenodo.8037209

Comments from the reviewer are given in black.

Author responses are given in red, and **amendments or additions made to the revised manuscript in bold red.**

**RC3: 'Comment on tc-2022-265', Tian Li**

**General Comments**

In this research, Bryony Freer and co-authors mapped the short-term variations in grounding line locations at the Bungenstockrücken Ice Plain using ICESat-2 satellite laser altimetry with a new approach of calculating elevation anomalies. They observed > 15 km tidal GL migration and discovered four different modes of tidal GL migration which are useful in validating numerical model simulations of GL migration and understanding the tidal processes of the ice shelf-ocean-subglacial system. Overall, I find this study interesting, robust and provide new insights into tidal GL migrations, which is important in determining long-term GL changes. The paper is well written and the topic fits well in this journal. I have several moderate concerns detailed below and I hope the authors could address them in the revised version.

We would like to thank you for your detailed comments and express our appreciation in particular for the deep knowledge and expertise that you have brought to this review. It has greatly helped to improve the manuscript.

**Major Comments**
**Inaccurate statements on mean-profile method:**

In Section 3.2.2, the authors claim that the mean elevation profile approach cannot be used to calculate the fine-scale short-term GL migrations with ocean tides compared to the newly proposed "Lowest-Sampled Tide" (LST) approach, this is not technically correct.

First, it does not make sense to map short-term GL migrations using a mean elevation profile of all 7 repeat cycles (in the case of RGT 559 GT3L). The nature of using an average elevation profile of all repeat cycles across the study period has already determined that we can only derive one Point F - which is the most landward ice flexure location experienced by the ice shelf among all the repeat cycles. This is the reason you see the clustered Point F features in Figure 2c. But this does not mean that the mean elevation profile method itself cannot be used to derive fine-scale short-term GZ features, which can be achieved by reducing the number of cycles used in calculating the elevation anomalies using a mean elevation profile.

Second, as the authors already mentioned "only two repeat measurements are required to locate Point F using RTLA" in Line 131, if you iterate different combinations of any two ICESat2 repeat cycles, calculate the mean elevation of these two cycles and estimate Point F from elevation anomalies based on this mean elevation profile, you will get a very detailed spatial sampling of the short-term Point F locations along one ground track, similar to the results in Figure 2e. For example, using only cycles 9 and 11 along RGT559 GT3L with the mean elevation approach can produce a most seaward Point F location between -80.90 and – 80.85 latitudes.

Many thanks for these comments; we agree that the manuscript will be strengthened by clarifying a number of the points that you make here.

To address your first point, our aim in discussing the mean-profile method here is to demonstrate that the 'traditional' RTLA method (using a single mean reference profile calculated from all available cycles, to locate a single Point F per track) cannot directly be used to study tidal migration of Point F - as you have pointed out. Indeed, this 'traditional' approach could be modified by reducing the number of cycles used to calculate the reference profile (as suggested in lines 241-242), although wherever >2 cycles are used to calculate a mean reference profile in an area subject to tidal GL migration, the interpretation of the resulting elevation anomalies per cycle (and therefore the location of Point F) will always be ambiguous.

As you have suggested, if we were to reduce the number of cycles used to calculate the mean profile down to 2, it would be possible to achieve similar sampling of tidal GZ processes. In our view this is a separate approach to the 'traditional' method for defining the reference profile that we were comparing to in Section 3.2.2 and Figure 2 (as it involves calculating several reference profiles for each track). This addresses why it hadn't initially been directly mentioned in Section 3.2.2, but following your comment we appreciate that this is unclear. **Therefore, in the revised manuscript we discuss the possibility of this alternative approach applying a mean reference profile using combinations of pairs of repeat cycles (Section 3.2.2).**

However, we remain confident that the LST method is a more effective approach. We discuss our reasoning for this in further detail below.

To address your second point, it is true that if we calculate along-track elevation anomalies using just 2 cycles, we should locate the same Point F position regardless of whether we use a reference profile calculated from *(a)* the mean of those two cycles, or *(b)* the elevation profile of the lower of the two sampled tides. This is illustrated in Figure R1. It is worth noting that in either case, it is only possible to locate Point F for the cycle sampled at the *higher* tide – i.e. For RGT 559, the derived Point F is attributed to the inland limit of tidal ice shelf flexure measured at Cycle 4; we cannot locate Point F for Cycle 9 in either case. Nevertheless, if we are reducing the number of cycles used to calculate elevation anomalies down to 2 anyway, we propose that it makes most sense to use the LST as the reference profile for a number of reasons:

1. It makes more sense conceptually. In using the LST as reference, we are comparing to a 'real' ice shelf surface profile, which gives a more realistic representation of the impact of tidal forcing between cycles on ice shelf flexure.
2. It provides a consistent reference profile, which means we can directly compare the derived Point F locations across anomaly profiles of all available cycles together along the same ground track – as shown in Figure 2(e). This would not be possible if you are recalculating a different mean reference profile for each combination of cycles.
3. It gives us a larger signal in the elevation anomalies, with a stronger change in gradient at Point F. This improves the success of the method use to locate point F using peaks in the $2^{nd}$ derivative of the elevation anomaly (as described in Section 3.2.3).
4. It minimises the required calculations in the processing chain, as only one reference profile has to be defined (the LST), instead of re-calculating a mean profile for each combination of cycles.

For these reasons, we are confident in our approach using the LST that has been presented in this paper. **For additional clarification we have outlined these advantages in section 3.2.2 of the revised manuscript. Within the remit of this paper, we think that this sufficiently justifies our choice of approach used to produce the results presented in the study.**

[Figure]

***Figure R1:*** *Comparing the derived Point F location on RGT 559 GT3L using just cycles 4 and 9 (highest and lowest sampled tides), using the mean reference profile (upper panel) vs the lowest sampled tide profile as reference (lower panel).*

In fact, the mean elevation approach has several benefits over the LST approach proposed in this study:

1) **Denser spatial sampling:** the number of different repeat cycle combinations available for Point F calculation will be higher than LST, because the latter only compares different repeat cycles to one fixed cycle with the lowest-sampled tide.

As discussed above, regardless of the combination of repeat cycles used to calculate a mean reference profile, you will only be able to locate Point F for the cycle with the higher sampled tide. For example, there would be no added benefit (and no improvement on spatial sampling) if you were to calculate elevation anomalies along RGT 559 GT3L using mean reference profiles for cycles 4+6, 4+8, 4+9, 4+11, 4+12 and 4+13 (Figure 2), because you would still only locate the same single Point F position for cycle 4 (the highest tide).

**For this reason, we have not included this in the revised manuscript.**

2) **Higher temporal resolution:** using the neighboring two repeat cycles can provide Point F with a time resolution of 91 day, this cannot be achieved by LST.

This is true in terms of absolute temporal resolution between measurements, which is particularly relevant when looking at long-term GL position change. However, when investigating tidal GL migration patterns in a stable GZ region like Bungenstockrücken (i.e. not experiencing significant long-term advance or retreat), the temporal resolution can be understood more as the range of the overall tide cycle that has been sampled. Tidal changes are inherently sub-daily; for example, there is effectively a 6-hour time difference between the measurement of Point F at high tide vs low tide. On tidal timescales the difference between using a mean vs LST approach therefore makes no difference.

**For this reason, we have not included this in the revised manuscript.**

3) **Free from errors in tidal models:** the mean elevation approach doesn't need tidal model to determine a reference profile.

The LST method also does not rely on tide model input at this stage. The cycle used as the lowest sampled tide is determined using the elevation anomaly measurements alone, with the modelled tide data just used for reference here.

**This has been clarified in the revised manuscript: lines 155 and 240-241.**

Third, in contrast to the statement in **Line 227-229** *"This also overcomes the issue of the mean reference profile being skewed by inland flexure at higher tides, but with the additional advantage that it can be applied consistently across RGTs, enabling automation"*, the automation of the iteration of different repeat-cycle combinations with the mean elevation approach is straightforward to implement, actually the method should be similar to the LST method shown in Figure 3.

The statement about automation refers to the specific advantage compared to using a *neutral* tide profile as reference, as opposed to the *mean* reference profile approach. **The wording has now been clarified in the revised manuscript: lines 243-246.**

In addition, using only two cycles can also overcome the issues of skewed ice flexure at high tide.

As discussed above, this was only in reference to the 'traditional' method using a mean reference profile calculated from several cycles. **This has been clarified in the revised manuscript: line 244.**

Therefore, I would like to hear the authors' opinion on this point, and would like to see a comparison between Point F locations calculated from any given two repeat cycles using the mean elevation approach and the results from their LST method.

**The comparison between Point F locations is given in Figure R1 and has been discussed in detail above.**

**We have given our opinions on this wider point, and considering the updates made to section 3.2.2 (discussed above) we do not think it is necessary to include Figure R1 or more detailed comparison of all possible methods in the final manuscript.**

**Tidal model accuracy:**

The Neutral Tide and the LST methods proposed in this study rely on tidal model predictions from CATS2008. Previous research show that the ice at deep grounding zone at narrow ice shelf embayment may not respond adequately in phase with ocean tides (Li et al., 2023; Reeh et al., 2000), which means that tidal model cannot always provide accurate tidal amplitude predictions. If this is the case, will the proposed approach still be valid for deriving short-term Point F locations and how the inaccurate tidal model predictions will impact your Point F result? Please clarify.

As stated above (and now clarified in the revised manuscript), the LST method does not directly use the output from the tide model to locate Point F, so this will not impact the results in any way.

In the case of using the Neutral Tide method, the possible model inaccuracies closer to the GL would need to be taken into account in such settings, as we would want to ensure that the cycle used as a

'neutral tide' is in fact as close to a 0m neap tide. However, as discussed in the paper, the Neutral Tide method is in general not suitable for automation and application across multiple tracks and so we don't apply it in our study. **To address this comment, we have added clarification in the revised manuscript: lines 235-238.**

**Specific Comments**

**Line 37:** Please rephrase this sentence, GZ can be wider than 10 km as demonstrated by the results of >15 km migration in this study.

**This has been rephrased in the revised manuscript.**

**Line 45-50:** This paragraph lacks mentioning previous research in studying the short-term GL migrations using satellite altimetry and DInSAR, such as:

- Brunt et al. (2011): https://www.cambridge.org/core/journals/journal-ofglaciology/article/analysis-of-ice-plains-of-the-filchnerronne-ice-shelf-antarctica-usingicesat-laser-altimetry/80F41E7FDC8124136AF50615861D5C71
- Milillo et al. (2017): http://doi.wiley.com/10.1002/2017GL074320
- Milillo et al. (2019) https://www.science.org/doi/10.1126/sciadv.aau3433
- Brancato et al. (2020): https://onlinelibrary.wiley.com/doi/10.1029/2019GL086291
- Milillo et al. (2022): https://www.nature.com/articles/s41561-021-00877-z

Brunt et al. (2011), Milillo et al. (2017) and Brancato et al. (2020) are already cited, but **we have added references to the other suggested papers in the revised manuscript here:**

- **Milillo et al. (2022) – Lines 43, 554**
- **Milillo et al. (2019) – Lines 43, 63, 583, 618**
- **Brancato et al. (2020) – additional reference added in line 118**

**Line 52-53:** What are the spatial and temporal resolutions of these historical archives? What is the requirement of spatial-temporal resolution for assessing the tidal GL migration?

These are currently discussed in more detail in section 2.2. The question of temporal resolution has been addressed in the reply to your major comments, but the main requirement is that we sample the ice shelf surface elevation at a range of different tides within a period over which there has not been significant long-term GL change. This will vary across different regions depending on the rate of long-term GL change and the extent of tidal GL migration, which is not well defined in most places.

In terms of spatial resolution, this is perhaps more to do with lack of spatial coverage in historical records, including lack of coherent SAR images for DInSAR methods, missing data at high latitude GZs, and the wider track spacing (and sometimes misaligned repeat-tracks) of ICESat.

**This has been updated to *'… with sufficient spatial coverage and temporal resolution'*: lines 50-51.**

**Line 63-65:** "where short-term GL migration is likely to impact both ice dynamics through rapid variations in basal shear stress, and basal melt rate through changes in cavity geometry enhancing tidal mixing." Here needs a reference.

**We have added references to Chen et al. (2023), Ciracì et al. (2023) and Milillo et al. (2019) to support this statement in the revised manuscript: line 63.**

**Line 109-110:** Dawson and Bamber (2020) also mapped Point H in addition to Point F in their study.

Thank you for this suggestion, **this has been updated in the revised manuscript: lines 107-8.**

**Line 111:** Need to mention why CryoSat-2 is not suitable to detect short-term GL changes – due to its long repeat cycle (369 day) (Dawson and Bamber, 2017).

**This has been added to the revised manuscript as suggested: lines 108-9.**

**Line 116-122:** Mohajerani et al. (2021) have also improved the DInSAR sampling under different tidal phases using Sentinel-1a/b SAR images across the Antarctic Ice Sheet.

**This has been added to the revised text, as suggested, alongside a reference the recent publication of Chen et al. (2023) who also employ Sentinel-1 DInSAR to detect tidal GL migration: lines 114-5.**

**Line 128:** Please mention that the "unrecoverable topographic biases across GZs" is caused by the large across-track slope across the GZ.

**This has been added to the revised manuscript: line 127.**

**Line 128-131:** *"In contrast, the Advanced Topographic Laser Altimeter System (ATLAS) that launched on board ICESat-2 in 2018 has a six-beam design with more accurate pointing, which reduces across-track deviation from the reference ground track (RGT), providing better spatial sampling of the GZ."* Not accurate and need clarification:

1) The six-beam design reduces the across-track deviation from the Reference Pair Track (RPT) inside each beam pair, not the Reference Ground Track (See https://nsidc.org/sites/default/files/icesat2_atl06_atbd_r005.pdf).
2) Please give a number of the ICESat-2 point control accuracy here and how it compares to ICESat. Luthcke et al. (2021) show that the performance of ICESat-2 can achieve 4.4 ± 6.0 m, this represents a very small across-track separation across repeat cycles.

**To address both of these points, the text has been revised in lines 128-130 to:**

*'In contrast, the Advanced Topographic Laser Altimeter System (ATLAS) that launched on board ICESat-2 in 2018 has a six-beam design with more accurate pointing, which reduces across-track deviation from each reference pair track (RPT) to within 4.4 ± 6.0 m (Luthcke et al., 2021), providing better spatial sampling of the GZ.'*

(Note, we therefore redefine RGT in section 3.1, where it now appears for the first time).

**Line 133:** Here I suggest the authors to add some discussions on the pros and cons of Li et al. (2022b)'s results, and talk about why it is necessary to extend the data record in this study.

**To address this, the following text has been added to the revised manuscript in lines 133-137:**

*'This is a highly valuable and comprehensive dataset for the identification of long-term grounding line change, but as only a single Point F has been located per ICESat-2 ground track, it is limited in use for studying tidal GZ processes. Therefore, here we extend the Li et al. (2022c) record at Bungenstockrücken to locate multiple Point F positions along each ground track as it migrates over the tide cycle (Fig. 1b), providing novel observations of tidal GZ behaviour.'*

**Line 142-144:** I think there is no need to mention the switch between weak and strong beams because this study did not use this feature in GZ calculation.

**This has been removed in the revised version, as suggested.**

**Line 149-151:** see my previous comment, can also merge this sentence into Line 142-144.

**This has been updated in the revised version, as suggested.**

**Line 152:** *"We obtained coincident tide amplitudes at the most seaward point of each ICESat2 ground track per cycle".* Please provide an average distance between these ICESat-2 seaward points used in the tidal amplitude calculation and a reference (historic) grounding line. This is important because if the seaward point still locates inside the GZ, then the modelled tidal amplitude may not represent the actual ice shelf elevation change in hydrostatic equilibrium. In addition, the orientation of the ICESat-2 grounds tracks are not always perpendicular with the actual grounding line, this can also introduce biases (Li et al., 2020, 2022a).

Thank you for raising this point. We were careful to ensure that all locations where modelled tides were extracted were further seaward than the known Point H, precisely to avoid this issue of the ice within the GZ not being hydrostatic equilibrium. However, as discussed above, these modelled tides were just used for reference and had no direct influence on the calculation of Point F locations.

**In response to this, we have added clarification in the revised manuscript that all tides were calculated at locations seaward of Point H, and clarify that these modelled tides were just used for reference: lines 155 and 159-60.**

**Line 157:** Please clarify why the annual tidal distribution is essential in GZ calculation.

This is already addressed briefly in lines 244-248, but **we have clarified this further in the revised manuscript: lines 157-159.** We provide a more detailed explanation below to explain our reasoning:

Again, as discussed above, the modelled tides are not used directly in the calculation of Point F location. Nevertheless, calculating the annual tide distribution is very important to give context for

the range of tides (and therefore Point F locations) that have been sampled by ICESat-2 over the study period. This is illustrated in Figure 2(f).

As the most extreme high and low tides occur less frequently, this helps us to understand how often the region experiences tidal forcing outside the range of tides sampled by ICESat-2. For example, as illustrated in Figure 2(f), RGT 559 has not been sampled by ICESat-2 during both the lowest 3% and highest 6.4% of tides experienced in the region. This indicates that 6.4% of the time, Point F could be located further inland than the maximum inland extent we have measured in this study. In order to determine the full width of the zone of ephemeral grounding, we would have to rely on ICESat-2 overpasses coinciding almost exactly with both the highest and lowest spring tides (each only occurring fortnightly).

**Line 172:** "minimum segment difference exceeds 1", I assume this is 1 meter, please add a unit.

**Yes, thank you for identifying this error. We have updated this in the revised manuscript: line 175.**

**Line 192-195:** Here the authors merely mention the traditional mean elevation approach in calculating elevation anomalies, then directly propose two new methods without any discussions on the disadvantages of the mean profile method. It is not clear why it is essential to develop two new approaches, why mean elevation approach fails to map short-term GL locations, and how the two new methods can further improve these research gaps. I suggest incorporating some of the information in **Line 201-235** to the beginning of **Section 3.2.2.**

**This has now been addressed in the response to your major comments above, with clarifications added in the section 3.2.2 of the revised manuscript. Whilst we have considered your suggestion to merge information from line 201-235 into the beginning of section 3.2.2, we believe that it would disrupt the flow of the text, so we have chosen to keep with the original structure.**

**Line 201-213:** As I have mentioned in the major comment, mean profile approach can provide high spatial-temporal resolution GZ calculation, please consider modify the relevant content accordingly.

This has been addressed in the response to your major comments above.

**Line 229-230:** The mean elevation profile approach can allow the calculation of Point F from repeat cycles both sampled at low tides, please see my major comments, and rephrase this sentence.

The major comments have been addressed above. **To clarify this point, we have rephrased the text in lines 243-4 to:**

**_'This also overcomes the issue of the mean reference profile (calculated from multiple cycles of data) being skewed by inland flexure at higher tides, …'_**

**Line 500-501:** _"The 15 km observed tidal GL migration at Bungenstockrücken is much larger than standard model grid spacing at the GL"_, what is the standard model grid size at the GL?

**We have added clarification of this in the revised manuscript, lines 525-527:**

*"The 15 km observed tidal GL migration at Bungenstockrücken is much larger than standard model grid spacing at the GL, which is typically less than 2km, and down to 250m where mesh refinement is applied at the GL (Cornford et al., 2020)"*

**Line 500-503:** *"Similarly, the time step of an ice sheet model is generally much longer than one day; therefore, prescribing a sub-daily change in GL position is not generally possible."* Not sure why mentioning sub-daily GL change here, especially the results in this paper cannot achieve the sub-daily resolution either.

Tidal changes are inherently sub-daily; by measuring how Point F migrates at different stages of the tide cycle we are therefore effectively measuring how the GL migrates within sub-daily timescales, even when the absolute repeat time between measurements ranges from 91 days to several years. This is made possible by the fact that this is a region with long-term stability and so we can assume that the majority of the GL migration is driven by tidal ice shelf flexure. This is confirmed by comparing the modelled ice shelf surface anomalies with the modelled tide heights (e.g. in Figure 2c-e). We mention this at this point in the Discussion in the context of ice shelf modelling, to highlight the importance of considering the possible impact of these sub-daily tidal processes on wider ice shelf and ice sheet dynamics.

**Following this justification, we do not think it is necessary to remove the discussion of sub-daily GL position change here.**

**Figures 2c,d,e:**

I assume the reference elevation profile of each different approach is plotted in the upper panel of subfigures c/d/e, however it is very difficult to discern them from the ICESat-2 repeat profiles. Please change the line symbols of reference elevation profile under different scenarios. Here are some suggestions:

1) consider only include one subfigure to show all the ICESat-2 repeat cycles with a colormap.
2) then add a different subfigure, plot three different reference elevation profiles in distinct colors or symbols and add all ICESat-2 repeat cycles in light grey color with low transparency as a background - this is to make sure that the reference elevation profiles can stand out from ICESat-2 profiles and the readers can clearly distinguish these three different reference profiles.

Thank you very much for these useful suggestions. We have experimented with a few different amendments to Figure 2 to improve the readability of the elevation profile panels. We propose the most effective solution is to include a smaller range of elevation values close to the ice shelf surface, which better highlights the surface profiles of each repeat cycle. By reducing the opacity of these repeat profiles and increasing the line thickness of the reference profile in each scenario, this shows the difference between the three reference profiles more effectively.

**We have amended Figure 2 (c-e) in the revised manuscript accordingly. The update to these panels is shown here in Figure R2.**

[Figure]

*Figure R2: Proposed amendment to panels c-e in Figure 2 to improve the readability of the reference elevation profiles.*

**Figure 6:** Both panels f and l are RGT 1223 1R, why do the Point F locations along the same ground track behave so differently? Is this a labelling error in the figure?

Yes, this is a labelling error – many thanks for spotting this. **Panel (l) has been updated in the revised manuscript to RGT 1138 1R.**

**Additional References included in the Revised Manuscript**

Brancato, V., Rignot, E., Milillo, P., Morlighem, M., Mouginot, J., An, L., Scheuchl, B., Jeong, S., Rizzoli, P., Bueso Bello, J. L., and Prats-Iraola, P.: Grounding Line Retreat of Denman Glacier, East Antarctica, Measured With COSMO-SkyMed Radar Interferometry Data, Geophys. Res. Lett., 47, https://doi.org/10.1029/2019GL086291, 2020.

Chen, H., Rignot, E., Scheuchl, B., and Ehrenfeucht, S.: Grounding Zone of Amery Ice Shelf, Antarctica, From Differential Synthetic-Aperture Radar Interferometry, Geophysical Research Letters, 50, e2022GL102430, https://doi.org/10.1029/2022GL102430, 2023.

Ciracì, E., Rignot, E., Scheuchl, B., Tolpekin, V., Wollersheim, M., An, L., Milillo, P., Bueso-Bello, J.-L., Rizzoli, P., and Dini, L.: Melt rates in the kilometer-size grounding zone of Petermann Glacier, Greenland, before and during a retreat, Proc. Natl. Acad. Sci. U.S.A., 120, e2220924120, https://doi.org/10.1073/pnas.2220924120, 2023.

Cornford, S. L., Seroussi, H., Asay-Davis, X. S., Gudmundsson, G. H., Arthern, R., Borstad, C., Christmann, J., Dias Dos Santos, T., Feldmann, J., Goldberg, D., Hoffman, M. J., Humbert, A., Kleiner, T., Leguy, G., Lipscomb, W. H., Merino, N., Durand, G., Morlighem, M., Pollard, D., Rückamp, M., Williams, C. R., and Yu, H.: Results of the third Marine Ice Sheet Model Intercomparison Project (MISMIP+), The Cryosphere, 14, 2283–2301, https://doi.org/10.5194/tc-14-2283-2020, 2020.

Luthcke, S. B., Thomas, T. C., Pennington, T. A., Rebold, T. W., Nicholas, J. B., Rowlands, D. D., Gardner, A. S., and Bae, S.: ICESat-2 Pointing Calibration and Geolocation Performance, Earth and Space Science, 8, https://doi.org/10.1029/2020EA001494, 2021.

Milillo, P., Rignot, E., Rizzoli, P., Scheuchl, B., Mouginot, J., Bueso-Bello, J., and Prats-Iraola, P.: Heterogeneous retreat and ice melt of Thwaites Glacier, West Antarctica, Sci. Adv., 5, eaau3433, https://doi.org/10.1126/sciadv.aau3433, 2019.

Milillo, P., Rignot, E., Rizzoli, P., Scheuchl, B., Mouginot, J., Bueso-Bello, J. L., Prats-Iraola, P., and Dini, L.: Rapid glacier retreat rates observed in West Antarctica, Nat. Geosci., 15, 48–53, https://doi.org/10.1038/s41561-021-00877-z, 2022.

Mohajerani, Y., Jeong, S., Scheuchl, B., Velicogna, I., Rignot, E., and Milillo, P.: Automatic delineation of glacier grounding lines in differential interferometric synthetic-aperture radar data using deep learning, Sci Rep, 11, 4992, https://doi.org/10.1038/s41598-021-84309-3, 2021.

Stubblefield, A. G., Spiegelman, M., and Creyts, T. T.: Variational formulation of marine ice-sheet and subglacial-lake grounding-line dynamics, J. Fluid Mech., 919, A23, https://doi.org/10.1017/jfm.2021.394, 2021.